# Rad53 checkpoint kinase regulation of DNA replication fork rate via Mrc1 phosphorylation

**Allison W McClure, John FX Diffley\***

Chromosome Replication Laboratory, The Francis Crick Institute, London, United Kingdom

**Abstract** The Rad53 DNA checkpoint protein kinase plays multiple roles in the budding yeast cell response to DNA replication stress. Key amongst these is its enigmatic role in safeguarding DNA replication forks. Using DNA replication reactions reconstituted with purified proteins, we show Rad53 phosphorylation of Sld3/7 or Dbf4-dependent kinase blocks replication initiation whilst phosphorylation of Mrc1 or Mcm10 slows elongation. Mrc1 phosphorylation is necessary and sufficient to slow replication forks in complete reactions; Mcm10 phosphorylation can also slow replication forks, but only in the absence of unphosphorylated Mrc1. Mrc1 stimulates the unwinding rate of the replicative helicase, CMG, and Rad53 phosphorylation of Mrc1 prevents this. We show that a phosphorylation-mimicking Mrc1 mutant cannot stimulate replication in vitro and partially rescues the sensitivity of a *rad53* null mutant to genotoxic stress in vivo. Our results show that Rad53 protects replication forks in part by antagonising Mrc1 stimulation of CMG unwinding.

**\*For correspondence:**
john.diffley@crick.ac.uk

**Competing interests:** The authors declare that no competing interests exist.

## Introduction

In response to DNA replication stress such as low nucleotide levels or DNA damage, a cascade of events is orchestrated by the DNA replication checkpoint to ensure genome protection. DNA replication stress is detected by proteins that activate the apical protein kinase Mec1 in *Saccharomyces cerevisiae* (ATR in humans) (*Pardo et al., 2017*; *Saldivar et al., 2017*). Mec1 then activates the effector protein kinase Rad53 through two mediator proteins, Rad9 and Mrc1. Active Rad53 coordinates a broad response to promote cell survival by regulating damage-dependent transcription, cell cycle, deoxyribonucleotide triphosphate (dNTP) levels, and replication origin firing (*Bastos de Oliveira et al., 2012*; *Krishnan et al., 2004*; *Paulovich and Hartwell, 1995*; *Santocanale and Diffley, 1998*; *Travesa et al., 2012*; *Zegerman and Diffley, 2010*; *Zhao et al., 1998*). In addition, Rad53 plays an essential role in stabilising stalled replication forks, allowing them to restart replication, and promoting replication through damaged templates (*Lopes et al., 2001*; *Tercero and Diffley, 2001*; *Tercero et al., 2003*).

Wild-type cells progress very slowly through S phase in response to the DNA damaging agent MMS, but much faster in *rad53* or *mec1* mutant cells (*Paulovich and Hartwell, 1995*). The slow S phase progression in wild-type cells is mainly due to checkpoint-dependent inhibition of origin firing (*Tercero et al., 2003*; *Zegerman and Diffley, 2010*). Rad53 inhibits origin firing through multiple, redundant phosphorylation events of two essential firing factors, Sld3 and Dbf4; non-phosphorylatable mutants of Sld3 and Dbf4, when combined, show the same fast progression through S phase as *rad53* mutants (*Zegerman and Diffley, 2010*). However, unlike *rad53* mutants, the non-phosphorylatable *sld3, dbf4* double mutant does not show enhanced sensitivity to replication stress consistent with the idea that regulation of fork stability, rather than origin firing by Rad53 is crucial for cell viability. The Rad53 targets involved in regulating replication fork stability are currently unclear, but several studies have implicated the Mec1-Rad53 checkpoint in replication fork slow-down in

response to replication stress suggesting there may be a link between replication fork rate and stability (*Bacal et al., 2018*; *Kumar and Huberman, 2009*; *Mutreja et al., 2018*; *Seiler et al., 2007*).

Mrc1 and its human counterpart, Claspin, were initially characterised as mediators of the replication checkpoint (*Alcasabas et al., 2001*; *Kumagai and Dunphy, 2000*). Mrc1 and a second mediator, Rad9, act redundantly in activating Rad53 after replication stress and the *mrc1*$^{17AQ}$ mutant, which cannot be phosphorylated by Mec1, does not mediate Rad53 activation. In addition to its role in Rad53 activation, Mrc1 has a genetically separable role in regulating replisome progression in the absence of DNA damage. *mrc1Δ* cells progress slowly through S phase whilst *mrc1*$^{17AQ}$ cells show normal S phase progression (*Osborn and Elledge, 2003*), and, conversely, cells with C-terminal truncations of Mrc1 that can still activate the checkpoint with near normal kinetics show slow S phase progression (*Naylor et al., 2009*). Mrc1, along with two associated proteins Csm3 and Tof1 (Tipin/Timeless in human cells), has also been shown to stimulate replication fork rates in vitro (*Lewis et al., 2017*; *Yeeles et al., 2017*). Mrc1 associates with replication forks in S phase and has contacts with multiple replisome components (*Bando et al., 2009*; *Baretić et al., 2020*; *Gambus et al., 2006*; *Katou et al., 2003*; *Komata et al., 2009*; *Lou et al., 2008*), but how Mrc1 regulates fork progression is unclear. Here, we show that the ability of Mrc1 to stimulate replication is inhibited by Rad53 phosphorylation, implicating Mrc1 as both a mediator and a target of the checkpoint.

## Results

### Rad53 inhibition of origin firing in vitro via Dbf4 and Sld3

To understand in molecular detail how Rad53 regulates DNA replication, we have exploited the reconstitution of DNA replication with purified budding yeast proteins. In these experiments, the MCM double hexamers were assembled onto a 10.6 kb plasmid DNA template, then phosphorylated with Dbf4-dependent kinase (DDK), and finally firing factors, DNA polymerases, and accessory factors were added to initiate DNA replication. We followed replication progression by separating the products on alkaline agarose gels to visualise incorporation of radiolabelled dCTP. Rad53 inhibits late origin firing in vivo by phosphorylating two substrates: Dbf4 and Sld3 (*Lopez-Mosqueda et al., 2010*; *Zegerman and Diffley, 2010*). To determine if their phosphorylation directly inhibits their ability to promote replication, we pre-phosphorylated each individually with Rad53 and added them to replication reactions. To do this, we used Rad53 and the kinase-dead Rad53 mutant (K227A, D339A) purified after expression in *E. coli*. Rad53 expressed in *E. coli* is hyper-phosphorylated as previously shown (*Gilbert et al., 2001*), whilst the kinase dead mutant is not (*Figure 1—figure supplement 1A*). As shown in *Figure 1A*, pre-incubation of DDK with ATP caused a small shift in Dbf4 mobility in SDS-PAGE even in the absence of Rad53, presumably reflecting autophosphorylation (*Francis et al., 2009*; *Kihara et al., 2000*). However, there was a further shift of Dbf4 in the presence of wild-type Rad53, which was not seen with the kinase dead Rad53. *Figure 1B* shows that DDK pre-phosphorylated with Rad53 was unable to promote replication (lane 3), while pre-incubation of DDK with ATP alone (lane 2) or with ATP and the kinase-dead Rad53 mutant (lane 4) had no effect on replication. Whilst this work was in progress, Abd Wahab and Remus also found that Rad53 could directly inhibit DDK action at origins (*Abd Wahab and Remus, 2020*).

To test whether Rad53 also inhibited Sld3, we took a similar approach by pre-incubating Sld3/7 with Rad53 prior to addition to the replication reaction. Similar to DDK, pre-incubation of Sld3/7 with Rad53 and ATP resulted in reduced mobility of Sld3 in SDS-PAGE (*Figure 1C*) and inhibition of replication (*Figure 1D*, *Figure 1—figure supplement 1B*). This Rad53 inhibition of Sld3/7 was dependent on its kinase activity because the kinase-defective mutant did not inhibit replication (lane 3, *Figure 1D*) and because pre-incubation without ATP did not inhibit replication (lane 5, *Figure 1D*). Moreover, preincubation of Sld3/7 and Rad53 separately did not inhibit replication (lane 4, *Figure 1D*). This also shows that Rad53 does not inhibit replication initiation when added together with the firing factors: presumably, replication initiates before Rad53 has time to phosphorylate and inhibit Sld3 and DDK. Taken together, these results show that phosphorylation of DDK or Sld3/7 can block initiation, consistent with previous results in vivo (*Lopez-Mosqueda et al., 2010*; *Zegerman and Diffley, 2010*).

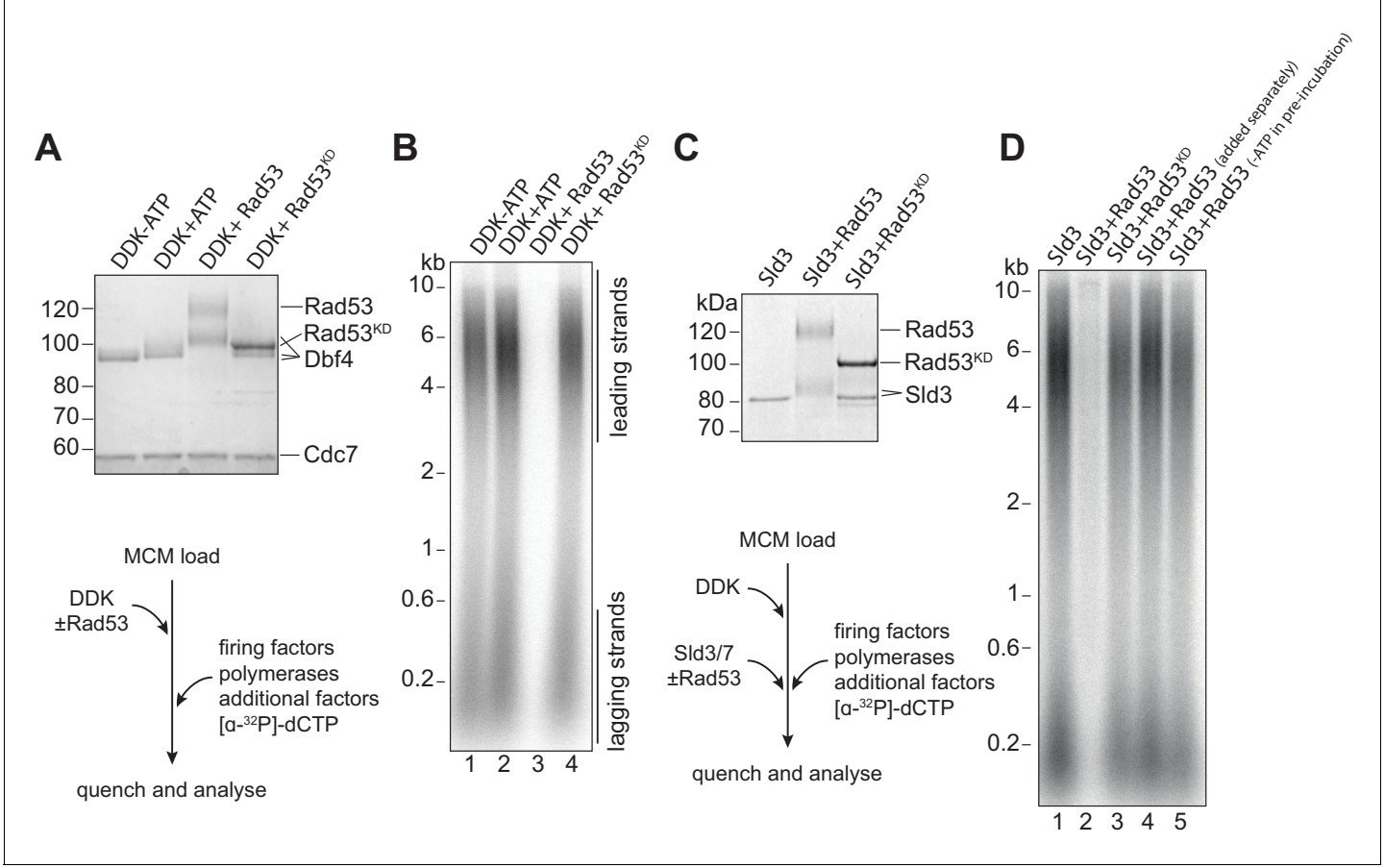

**Figure 1.** Rad53 inhibition of origin firing. (**A**) DDK was incubated with Rad53 or Rad53KD (K227A, K339A) for 15 min, separated by SDS-PAGE, and stained with coomassie. (**B**) DDK was incubated with Rad53 for 15 min and then added to a standard three-step in vitro replication reaction (see Materials and methods for more details). After 20 min, reactions were stopped with EDTA and products were separated on an alkaline agarose gel. (**C**) Sld3/7 was incubated with Rad53 as in (**A**). (**D**) Sld3/7 was incubated with Rad53 and added to a replication reaction. In lane 4, Sld3/7 and Rad53 were incubated separately from each other prior to addition to the replication reaction, and in lane 5, ATP was omitted during the pre-incubation. The online version of this article includes the following source data and figure supplement(s) for figure 1:

**Source data 1.** Original gel images for *Figure 1*.

**Figure supplement 1.** Rad53 phosphorylation of Sld3/7.

**Figure supplement 1—source data 1.** Original gel images for *Figure 1—figure supplement 1*.

## Rad53 inhibition of replication elongation via Mrc1 and Mcm10

Next, we wanted to determine whether Rad53 could affect replication elongation. We pre-incubated the elongation factor mix (RPA, Ctf4, TopoI, Csm3/Tof1, Mrc1, Polα, and Mcm10) with Rad53 and ATP, then added this to reactions after MCM loading, DDK phosphorylation, and firing factor addition. We stopped the reactions at early time points so that any effects of elongation could be more easily seen by the size of the leading strand replication products. *Figure 2A* shows that the sizes of leading strand products were reduced after Rad53 phosphorylation. *Figure 2B* shows that these reductions correspond to a decrease in replication fork rate from about 0.7 kb/min to 0.4 kb/min.

Mrc1 and Csm3/Tof1 (M/C/T) are non-essential proteins known to directly stimulate replication fork rate (*Lewis et al., 2017*; *Yeeles et al., 2017*). Their inactivation by Rad53, therefore, could provide an explanation for the reduction in fork rate caused by Rad53. Indeed, both Mrc1 and Csm3 (but not Tof1) exhibited reduced mobility in SDS-PAGE after incubation with Rad53 but not Rad53KD (*Figure 2C*), indicating that Rad53 can phosphorylate these proteins. When M/C/T were pre-incubated with wild-type Rad53, but not Rad53KD, leading strand product size after 7 min of replication was decreased (*Figure 2D*). This leading strand product size was decreased when Mrc1 was

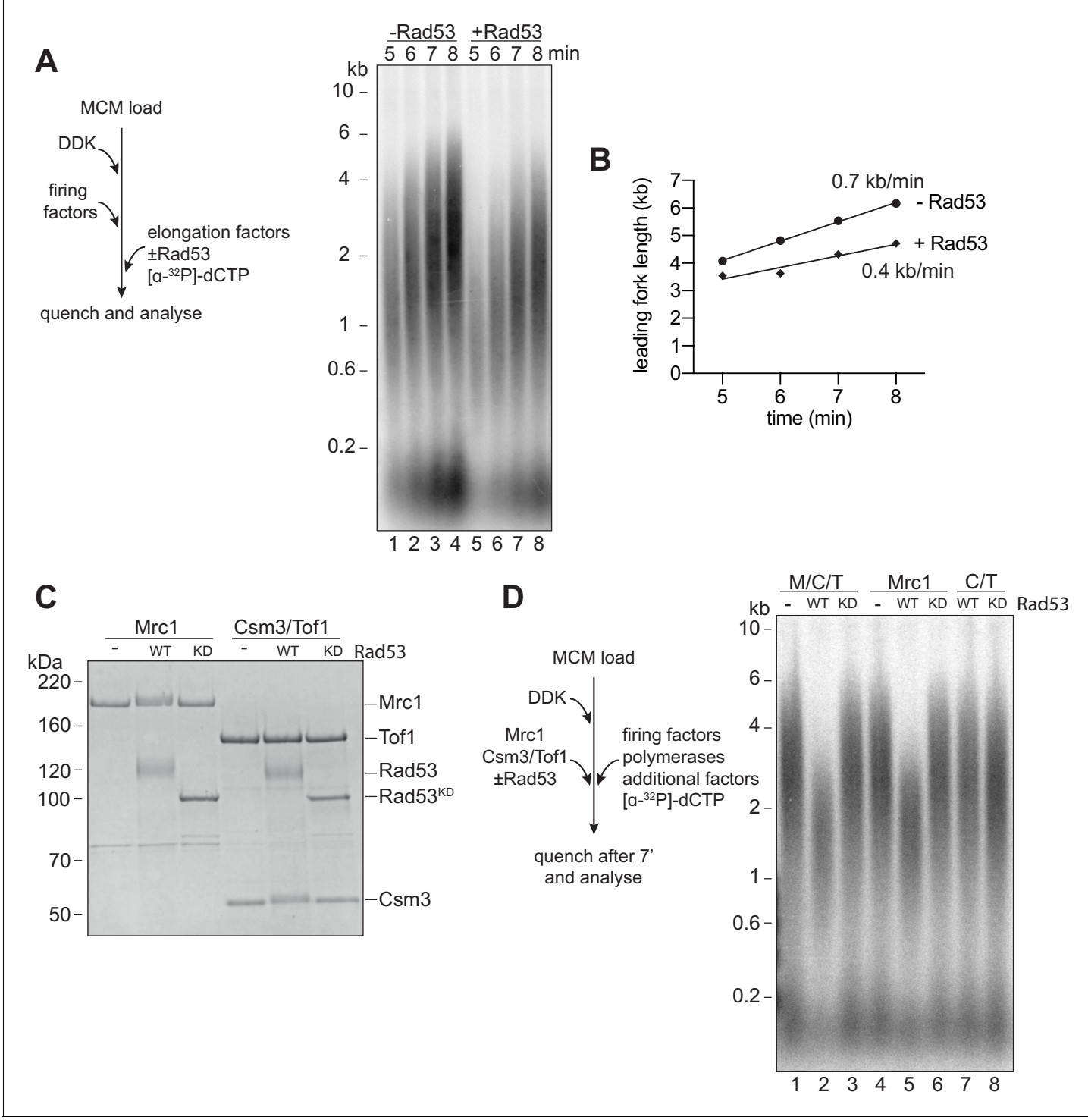

**Figure 2.** Rad53 inhibition of replication elongation via Mrc1. (**A**) Elongation factors (here, defined as RPA, Ctf4, TopoI, Mrc1, Csm3/Tof1, Polα, and Mcm10) were pre-incubated with Rad53 prior to addition to a four-step replication reaction that was stopped at the indicated timepoints. (**B**) Leading fork lengths (see Materials and methods for quantification method) at each timepoint from (**A**) with a linear fit. (**C**) Mrc1 or Csm3/Tof1 were incubated with Rad53, separated by SDS-PAGE, and stained with coomassie. (**D**) M/C/T or individual Mrc1 and Csm3/Tof1 were incubated with Rad53 prior to addition to a replication reaction. Reactions were stopped at 7 min (not completion).

The online version of this article includes the following source data and figure supplement(s) for figure 2:

**Source data 1.** Original gel images for *Figure 2*.

**Figure supplement 1.** Mrc1 and Rad53 binding.

*Figure 2 continued on next page*

*Figure 2 continued*

**Figure supplement 1—source data 1.** Original gel images for *Figure 2—figure supplement 1*.

incubated with Rad53 but not when Csm3/Tof1 was incubated with Rad53 (*Figure 2D*). These data indicate that phosphorylation of Mrc1 alone is sufficient to slow replication.

It has been suggested that Rad53 can inhibit one of its targets, DDK, by direct binding as well as by phosphorylation (*Abd Wahab and Remus, 2020*). We therefore asked whether we could detect stable interactions between Mrc1 and Rad53 by co-immunoprecipitation using the same buffer conditions used in the in vitro replication assay. After mixing Mrc1 and Csm3/Tof1, approximately equal amounts of Csm3/Tof1 and Mrc1 were detected in Mrc1 immunoprecipitates, consistent with the idea that Mrc1 forms a stable complex with Csm3/Tof1. Under these same conditions, however, little or no Rad53 or Rad53$^{KD}$ was detected in Mrc1 immunoprecipitates (*Figure 2—figure supplement 1A*). This, together with the fact that Rad53$^{KD}$ does not inhibit replication, suggests that phosphorylation of Mrc1 is likely to be most important for its inhibition.

Mrc1 has been shown to be phosphorylated by DDK kinase in fission yeast (*Matsumoto et al., 2017*). *Figure 2—figure supplement 1B* shows that the rate of replication in vitro was not affected after incubation of Mrc1 with either DDK or CDK together with ATP, suggesting that neither kinase can inhibit Mrc1 function in replication elongation.

To determine whether phosphorylation of Mrc1 is necessary for Rad53 to slow replication, we pre-incubated all of the elongation factors, except Mrc1, with Rad53 prior to replication. Addition of Mrc1 separately completely rescued the reduction in replication speed by Rad53 (*Figure 3A*). Together with the previous experiments, we conclude that Rad53 phosphorylation of Mrc1 is necessary and sufficient to explain the slowing of replication rate by Rad53.

It has recently been shown that Rad53 can inhibit the already slow rate of replication fork progression in the absence of M/C/T (*Devbhandari and Remus, 2020*), which led these authors to conclude that Rad53 inhibition of fork rate does not require M/C/T. To investigate this apparent discrepancy further, we pre-incubated the mixture of elongation factors lacking M/C/T (RPA, Ctf4, TopoI, Polα, and Mcm10) with Rad53. Consistent with this previous work (*Devbhandari and Remus, 2020*), the presence of Rad53 reduced replication rate in the absence of M/C/T (*Figure 3B*, compare lanes 7–9 with 10–12). However, when unphosphorylated M/C/T was added to the reaction separately from the pre-incubated elongation protein mix, replication speed was completely rescued to the rate seen in the absence of Rad53 (*Figure 3B*, lanes 13–15). From this we conclude that, in agreement with previous results, Rad53 can inhibit elongation via some target(s) other than M/C/T; however, this inhibition is only seen when unphosphorylated M/C/T is absent.

Mcm10 stimulates replication fork rate in the absence of Mrc1, but not in its presence (*Langston et al., 2017*; *Lõoke et al., 2017*), so inactivation of Mcm10 by Rad53 could explain our results. Incubation of Mcm10 with Rad53 reduced Mcm10 mobility in SDS-PAGE (*Figure 3C*) consistent with Rad53 phosphorylation of Mcm10. As shown in *Figure 3D*, incubation of Mcm10 with Rad53 reduced fork rate in the absence of Mrc1 (*Figure 3D* lanes 4 and 5) or in the presence of phosphorylated Mrc1 (*Figure 3D* lanes 2 and 3) but did not affect fork rate in the presence of unphosphorylated Mrc1 (*Figure 3E*). Interestingly, whilst pre-phosphorylation of Mcm10 with Rad53 affects its ability to accelerate fork rate, it did not inhibit replication initiation, suggesting that these functions of Mcm10 are separable.

## Mrc1 regulation of replication rate

How Mrc1 supports fast replication speed is not well understood. Mrc1 has many contacts with Polε (*Lou et al., 2008*), and Mrc1 stimulates DNA replication when Polε is synthesising the leading strand (*Figure 4A* and *Yeeles et al., 2017*). Mrc1 could, therefore, directly stimulate DNA synthesis by Polε (*Zhang et al., 2018*). We tested this idea by using a truncation of the catalytic domain of Polε (*Yeeles et al., 2017*; *Zhou et al., 2017*), which supports CMG formation but does not synthesise DNA. In these reactions, Polα is the only DNA polymerase, and Mrc1 still stimulates replication rate (*Figure 4B*). In addition, Mrc1 can stimulate DNA replication when Polδ is synthesising the leading strand (*Figure 4C* and *Yeeles et al., 2017*). Further, Rad53 inhibits Mrc1 stimulation of replication rate in all three conditions (*Figure 4A–C*). Therefore, Mrc1 can stimulate DNA synthesis regardless

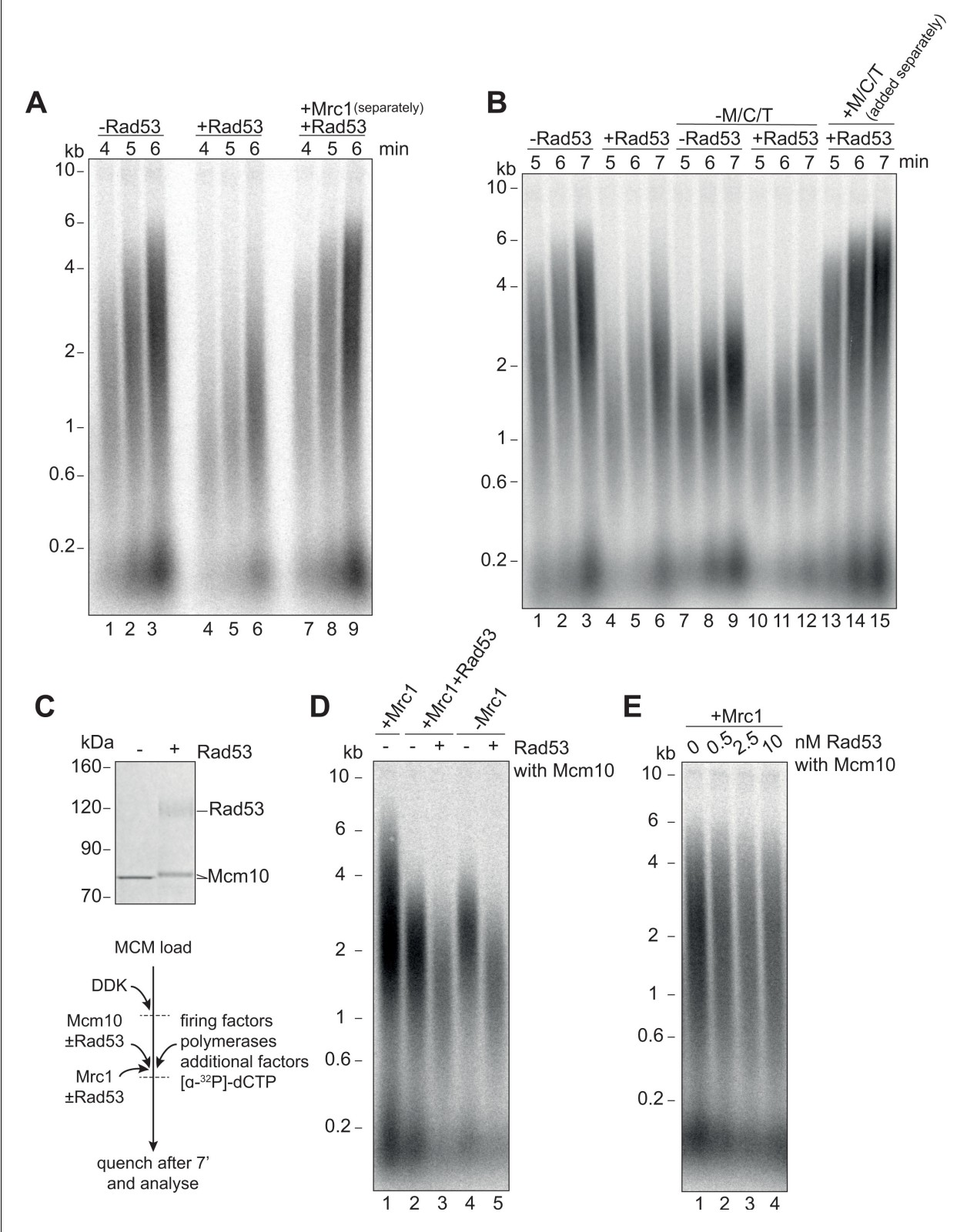

**Figure 3.** Mrc1 phosphorylation is necessary for Rad53 dependent inhibition of elongation. (A) Elongation factors were pre-incubated with Rad53 prior to addition to a four-step replication reaction that was stopped at the indicated timepoints. Mrc1 was omitted from the pre-incubation step in lanes 7–9 and added separately. (B) Reactions were performed as in (A) except M/C/T was omitted from the pre-incubation with Rad53 and added separately as indicated. (C) Mcm10 was incubated with Rad53, separated by SDS-PAGE, and stained with coomassie. (D) Mrc1 and Mcm10 were incubated with

*Figure 3 continued on next page*

*Figure 3 continued*

Rad53 prior to addition to a replication reaction (scheme to the left). Mrc1 was omitted from lanes 4 and 5. Reactions were stopped at 7 min. (**E**) Mcm10 was incubated with the indicated concentration of Rad53 prior to addition to a replication reaction in the presence of Mrc1.

The online version of this article includes the following source data for figure 3:

**Source data 1.** Original gel images for *Figure 3A and B*.
**Source data 2.** Original gel images for *Figure 3C, D and E*.

of which DNA polymerase is synthesising the leading strand. Moreover, Rad53 prevents Mrc1 stimulation regardless of which polymerase is synthesising the leading strand.

We next asked if Mrc1 can directly stimulate the activity of the CMG helicase. Using the appearance of an underwound form of circular plasmid (form U*) as a measure of CMG helicase activity, Devbhandari and Remus have recently shown that more U* product is generated in reactions containing M/C/T than in reactions lacking M/C/T (*Devbhandari and Remus, 2020*). This could indicate that M/C/T stimulates the rate of unwinding by CMG, or that M/C/T increases the ultimate extent of unwinding, or both. To distinguish these possibilities, we developed a new and quantitative assay to measure CMG activity (*Figure 4D*). Unwinding of double stranded DNA renders it resistant to cleavage by restriction endonucleases. The fraction of double- and single-stranded DNA can then be determined using qPCR with primers flanking restriction sites (*Zierhut and Diffley, 2008*). We constructed a linear DNA template with an origin of replication near one end and cassettes containing four tandem MseI restriction enzyme sites at 200 bp, 500 bp, 1000 bp, 1500 bp, and 2000 bp from the origin (*Figure 4D*). After loading the MCMs specifically at the origin (*Figure 4—figure supplement 1*), firing factors were added to form and activate CMG. Then, MseI restriction enzyme was added at indicated times for 3 min to cleave double-stranded DNA. The amount of unwound DNA is then measure by qPCR with primers flanking each of the cassettes. Using this assay, we found that ssDNA accumulated with time at each of the MseI sites, with a delay for distant sites from the origin reflecting time before CMG reached these sites (*Figure 4E*). We extracted from this data a rate of CMG helicase activity of ~79 bp/min (95% confidence interval, 71–88 bp/min). When we included Mrc1 in the reactions, ssDNA accumulated faster reflecting a CMG helicase rate of ~135 bp/min (95% confidence interval, 118–151 bp/min) (*Figure 4F*). This data shows that Mrc1 can directly increase the rate of CMG helicase unwinding. When we incubated Mrc1 with Rad53 prior to addition to the helicase reaction, ssDNA accumulated at a rate similar to the -Mrc1 reactions (*Figure 4G*), indicating that Rad53 inhibits Mrc1's ability to stimulate the CMG helicase.

## Identification of Rad53 phosphorylation sites in Mrc1

To understand how Rad53 regulates replication in vitro and in vivo, we undertook a mutational analysis of Mrc1 phosphorylation sites. The mutant Mrc1[17AQ] protein, which cannot be phosphorylated by Mec1 (*Osborn and Elledge, 2003*), supported normal replication speed and was inhibited by Rad53 just as the wild-type Mrc1 indicating that the Rad53 phosphorylation sites regulating Mrc1 activity in fork rate do not overlap the Mrc1 phosphorylation sites involved in signal transduction from Mec1 (*Figure 5A*). We note that the mobility of the Mrc1[17AQ] protein in SDS-PAGE was not reduced after incubation with Rad53 (*Figure 5C*), suggesting that Rad53 can phosphorylate at least one of these 17 S/T-Q sites despite the site/s not being relevant for Rad53-dependent inhibition.

Deletion of the C-terminus of Mrc1, which has been implicated in regulating S-phase progression in vivo, slowed replication rate in vitro, and was not further inhibited by incubation with Rad53 (*Figure 5B*). Furthermore, the mobility of the Mrc1 truncation mutant was not reduced in SDS-PAGE after incubation with Rad53 (*Figure 5C*). These results suggest that the C-terminal region of Mrc1 is important for its function in regulating fork rate, and key Rad53 phosphorylation sites regulating this function may lie in this region.

Rad53 phosphorylation sites cannot be reliably predicted by primary amino acid sequence, so we took three unbiased approaches to identify the sites in Mrc1 that can be phosphorylated by Rad53 in vitro. First, we expressed and purified five overlapping protein fragments of Mrc1 and incubated each in vitro with Rad53 and $\gamma^{32}$P-ATP. As shown in *Figure 5D*, the fragment containing the last 228 amino acids of Mrc1, which includes the region required for fork rate stimulation (*Figure 5A*), was the best substrate for Rad53 in vitro. Second, we incubated Rad53 and $\gamma^{32}$P-ATP with a peptide

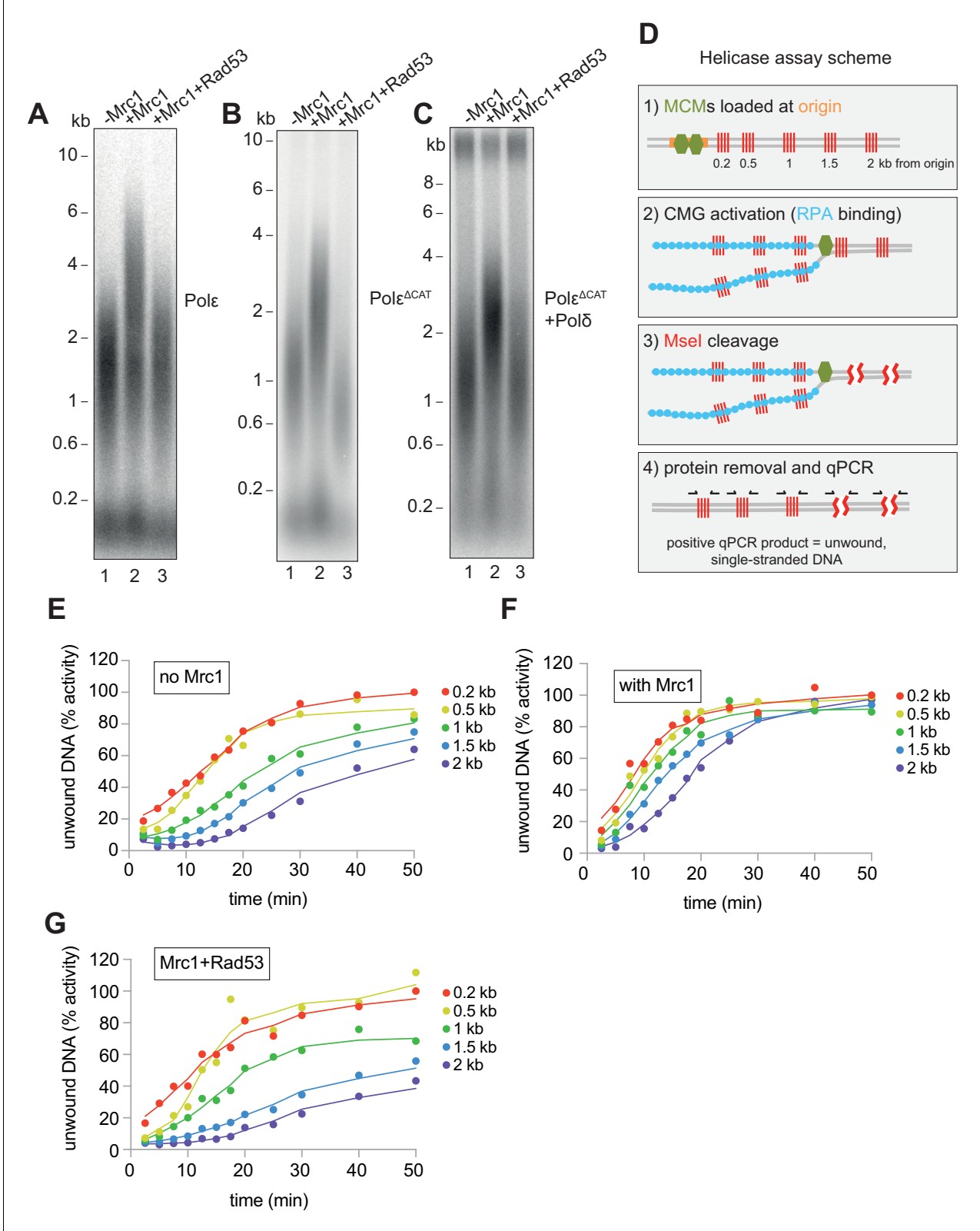

**Figure 4.** Mrc1 regulation of replication rate. (A) Mrc1 was omitted, incubated alone, or incubated with Rad53 prior to adding to a replication reaction. Reactions were stopped after 7 min. (B) Reactions as in (A) but using the catalytically-dead mutant (Polε$^{\Delta CAT}$) of Polε, and reactions were stopped at 10 min. (C) Reactions as in (A) but with Polε$^{\Delta CAT}$ and the addition of PCNA, RFC, and Polδ. (D) Helicase assay scheme. After MCMs are loaded specifically at the origin, CMGs are activated and unwind DNA. At each timepoint, MseI is added to digest DNA that is double-stranded; MseI does not digest

*Figure 4 continued on next page*

*Figure 4 continued*

single-stranded, RPA-coated DNA. The reactions are then quenched, proteins removed, and qPCR is performed using primers flanking the MseI cleavage sites, which generate a signal from the unwound DNA. (E) Timecourse of reactions depicted in (D). Data is normalised to the amount of unwound DNA at the closest MseI site (0.2 kb from the origin) at the last timepoint (see Materials and methods for more detail). (F) Reactions as in (E) with the addition of Mrc1. (D) Reactions as in (E) where Mrc1 is incubated with Rad53.

The online version of this article includes the following source data and figure supplement(s) for figure 4:

**Source data 1.** Original gel images for *Figure 4A and B*.
**Source data 2.** Original gel images for *Figure 4C*.
**Source data 3.** Source data for *Figure 4E, F and G*.
**Figure supplement 1.** Origin-specific MCM loading conditions.
**Figure supplement 1—source data 1.** Original gel images for *Figure 4—figure supplement 1*.

array on which the entire Mrc1 protein sequence was 'printed' as overlapping 20-mer peptides (*Figure 5—figure supplement 1A*). Many of the peptides that were phosphorylated by Rad53 in this experiment also mapped to the C-terminus of Mrc1. Lastly, we used mass spectrometry to identify amino acids specifically phosphorylated by Rad53 (*Figure 5—figure supplement 1B*). Consistent with the peptide array and fragment analysis, many of the phosphorylated residues were in the C-terminus of Mrc1.

We used this information to generate a non-phosphorylatable Mrc1 mutant in which serine and threonine residues were changed to alanine: such a mutant should promote faster replication after incubation with Rad53 than wild-type Mrc1. As shown in *Figure 5E*, we were able to generate mutants (Mrc1$^{14A}$ and Mrc1$^{19A}$) that indeed exhibited faster replication than wild-type Mrc1 after Rad53 phosphorylation (compare lanes 2, 4, and 6). Unfortunately, these mutants — especially Mrc1$^{19A}$ — did not stimulate replication to the rate of unphosphorylated wild-type Mrc1 (compare lanes 1, 4, and 6). This is likely due, in part, to the fact that these mutants exhibited defects in promoting faster replication in the absence of Rad53 indicating that they are not completely functional (compare lanes 1, 3, and 5). Moreover, even Mrc1$^{19A}$ was still inhibited slightly by incubation with Rad53 (compare lanes 5 and 6), suggesting that additional sites not mutated in this construct can still be phosphorylated by Rad53 and inhibit Mrc1 function. Cells expressing *MRC1$^{19A}$* as the sole copy of *MRC1* were not sensitive to the replication stress agent hydroxyurea (HU) (*Figure 5—figure supplement 1C*). Another mutant, Mrc1$^{41A}$ in which all the serines and threonines within the fragment 5 of *Figure 5D* were mutated to alanines was even more defective than the other mutants in the absence of Rad53 but was not further inhibited by Rad53 (*Figure 5—figure supplement 1D*). Thus, we were unable to generate an Mrc1 mutant with wild type function when unphosphorylated and completely resistant to inhibition by Rad53 phosphorylation.

## Mrc1$^{8D}$ slows fork rate in vitro and in vivo and partially rescues rad53 mutant sensitivity

As an alternative approach, we generated an Mrc1 mutant in which potential phosphorylation sites were replaced with aspartate, mimicking the negative charge of phosphate: such a mutant is predicted to be unable to promote faster replication even in the absence of Rad53. Indeed, a mutant in which eight serine/threonine Rad53 phosphorylation sites were changed to aspartate (Mrc1$^{8D}$) was unable to stimulate replication even in the absence of Rad53 (*Figure 6A*). These eight sites are a subset of the residues mutated in Mrc1$^{14A}$ mutant that still retains near wild-type activity when not incubated with Rad53 (*Figure 6A*). This suggests that the inactivation seen in Mrc1$^{8D}$ is not simply a consequence of changing essential serine or threonine residues. Three of the residues mutated in Mrc1$^{8D}$ (S911, S918, and S920) were identified in a phosphoproteomics screen of MMS-treated cells (*Lanz et al., 2021*), and another phosphoproteomics screen identified nearby residues that were dependent on the presence of Rad53 (*Lao et al., 2018*), suggesting that some of the sites in the Mrc1$^{8D}$ mutant are relevant to Rad53 activity in vivo.

Cells harbouring *MRC1$^{8D}$* as the only copy of *MRC1* were able to activate Rad53, evidenced by Rad53 hyperphosphorylation, at the same time and to the same extent as *MRC1$^+$* cells after release from a G1 arrest into HU, whilst *mrc1Δ* cells exhibited a reduced and delayed Rad53 activation (*Figure 6B*), consistent with previously published results (*Alcasabas et al., 2001*). To rule out any contribution of the other mediator in Rad53 activation, Rad9, we repeated this experiment with

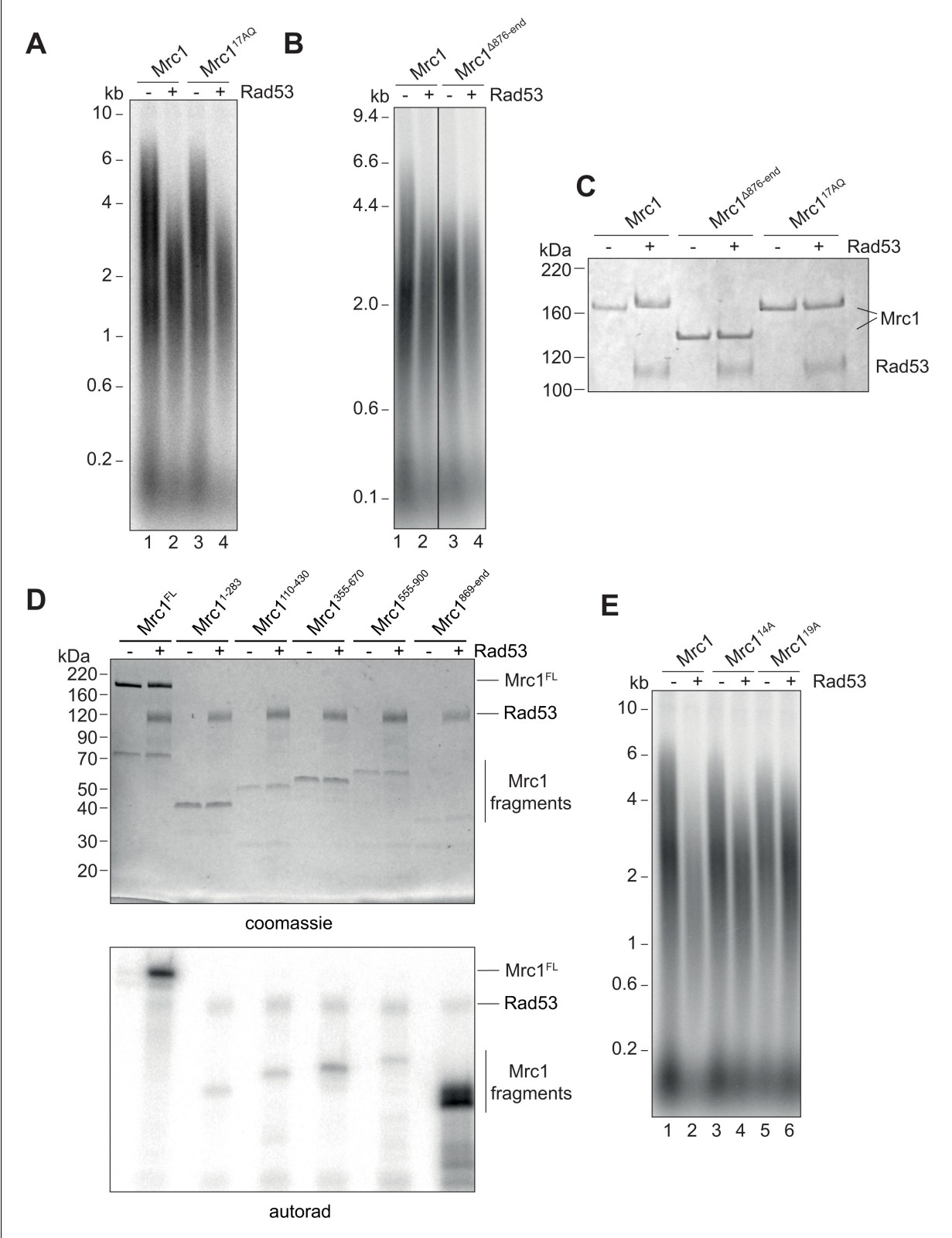

**Figure 5.** Identification of Rad53-dependent phospho-sites on Mrc1. (**A**) Mrc1$^{17AQ}$ mutant was incubated with Rad53 prior to addition to in vitro replication for 7 min. (**B**) C-terminal truncation of Mrc1 was incubated with Rad53 prior to addition to in vitro replication for 7 min. Note that all lanes were run on the same alkaline agarose gel, but other samples between lanes 2 and 3 were removed for clarity. (**C**) Mrc1, Mrc1 truncation, and Mrc1$^{17AQ}$ were incubated with Rad53 and separated by SDS-PAGE then stained with coomassie. (**D**) Fragments of Mrc1 were incubated with Rad53, separated by

*Figure 5 continued on next page*

*Figure 5 continued*

SDS-PAGE, then subjected to autoradiography. (E) Mrc1, Mrc1[14A], and Mrc1[19A] were incubated with Rad53 prior to addition to in vitro replication assay for 7 min.

The online version of this article includes the following source data and figure supplement(s) for figure 5:

**Source data 1.** Original gel images for *Figure 5A, B and C*.
**Source data 2.** Original gel images for *Figure 5D and E*.
**Figure supplement 1.** Identification of Rad53-dependent phospho-sites on Mrc1.
**Figure supplement 1—source data 1.** Original gel and plate images for *Figure 5—figure supplement 1*.
**Figure supplement 1—source data 2.** List of peptides synthesised for peptide array in *Figure 5—figure supplement 1A*.
**Figure supplement 1—source data 3.** Source data for *Figure 5—figure supplement 1B*.

strains in which *RAD9* was also deleted. As shown in *Figure 6C*, the *MRC1[8D]* mutant supported Rad53 phosphorylation at the same time and to the same extent as wild-type *MRC1*, even in the *rad9Δ* background (compare lanes 2–5, with lanes 7–10), whilst the *mrc1Δ*, *rad9Δ* double mutant was completely defective in Rad53 phosphorylation (lanes 12–15). *mrc1Δ rad9Δ* mutants are inviable (*Alcasabas et al., 2001*) and in this experiment were maintained by deletion of *SML1*; however, the *MRC1[8D] rad9Δ* double mutants were viable without deletion of *SML1*. Taken together, these results indicate that Mrc1[8D] can still signal from stalled replication forks, suggesting that the protein is stable in vivo (also seen in the Mrc1 immunoblots in *Figure 6B,C*) and maintains at least some interactions with the replisome in vivo.

To determine whether replication forks are slowed by the *MRC1[8D]* mutations in vivo, we performed a pulse-chase experiment. Briefly, cells expressing the human nucleoside transporter, hENT, and the *Drosophila melanogaster* deoxyribonucleoside kinase, DmdNK, (*Segurado and Diffley, 2008*; *Vernis, 2003*) from galactose-inducible promoters were first arrested in G1, then released into medium containing BrdU and HU to label and synchronise the replication forks from early-firing origins. Cells were then released into medium containing thymidine and lacking both HU and BrdU to allow elongation to resume and to prevent any further labelling with BrdU. Replication products were visualised after alkaline agarose gel electrophoresis and immunoblotting with an anti-BrdU antibody. *Figure 6D,E* shows that the *MRC1[8D]* mutant cells had shorter replication intermediates in HU and these intermediates were extended at a ~25% slower rate (*Figure 6D,E*) than *MRC1+* cells. The overall replication rate in *MRC1+* cells was slower than fork rates previously determined in budding yeast (*Hodgson et al., 2007*; *Sekedat et al., 2010*) probably because these cells were grown in galactose-containing medium at a reduced temperature (25°C). These results indicate that the *MRC1[8D]* mutant promotes slower fork rates than wild type *MRC1* in vivo as well as in vitro.

If one of the functions of Rad53 is to slow replication after replication stress, then the *MRC1[8D]* mutant might make *rad53Δ sml1Δ* cells more resistant to replication stress. Indeed, *MRC1[8D] rad53Δ sml1Δ* cells showed some improvement in survival, compared to *rad53Δ sml1Δ* cells when chronically exposed to low concentrations of either HU (2 mM) or MMS (0.006%), although they did not promote additional survival to higher concentrations (8 mM HU and 0.01% MMS) (*Figure 7A*). Similar results were obtained with two freshly germinated spores of each genotype (*Figure 7—figure supplement 1*) arguing that the suppression seen was not a result of suppressor mutations, which accumulate readily in *rad53Δ sml1Δ* cells (*Gómez-González et al., 2019*). *MRC1[8D]* also promoted increased survival in *rad53Δ sml1Δ* cells to acute exposure to higher concentration of MMS (0.02%) relative to *MRC1* wild type (*Figure 7B*). These data show that *MRC1[8D]* can partially rescue the sensitivity of *rad53Δ sml1Δ* cells in response to chronic and acute replication stress, suggesting that slowing replication forks may be part of Rad53's role in protecting replication forks during the replication checkpoint.

## Discussion

Our results show that, in addition to its role in checkpoint activation upstream of Rad53, Mrc1 also has a role downstream of the checkpoint, as a substrate of Rad53. Phosphorylation of Mrc1 by Rad53 prevents Mrc1-dependent stimulation of CMG unwinding, leading to a reduced replication fork rate. We suggest that linking these two roles could allow Rad53 to slow replication speed specifically at a stressed or damaged fork, and, therefore, act efficiently at stochastic fork stalling events

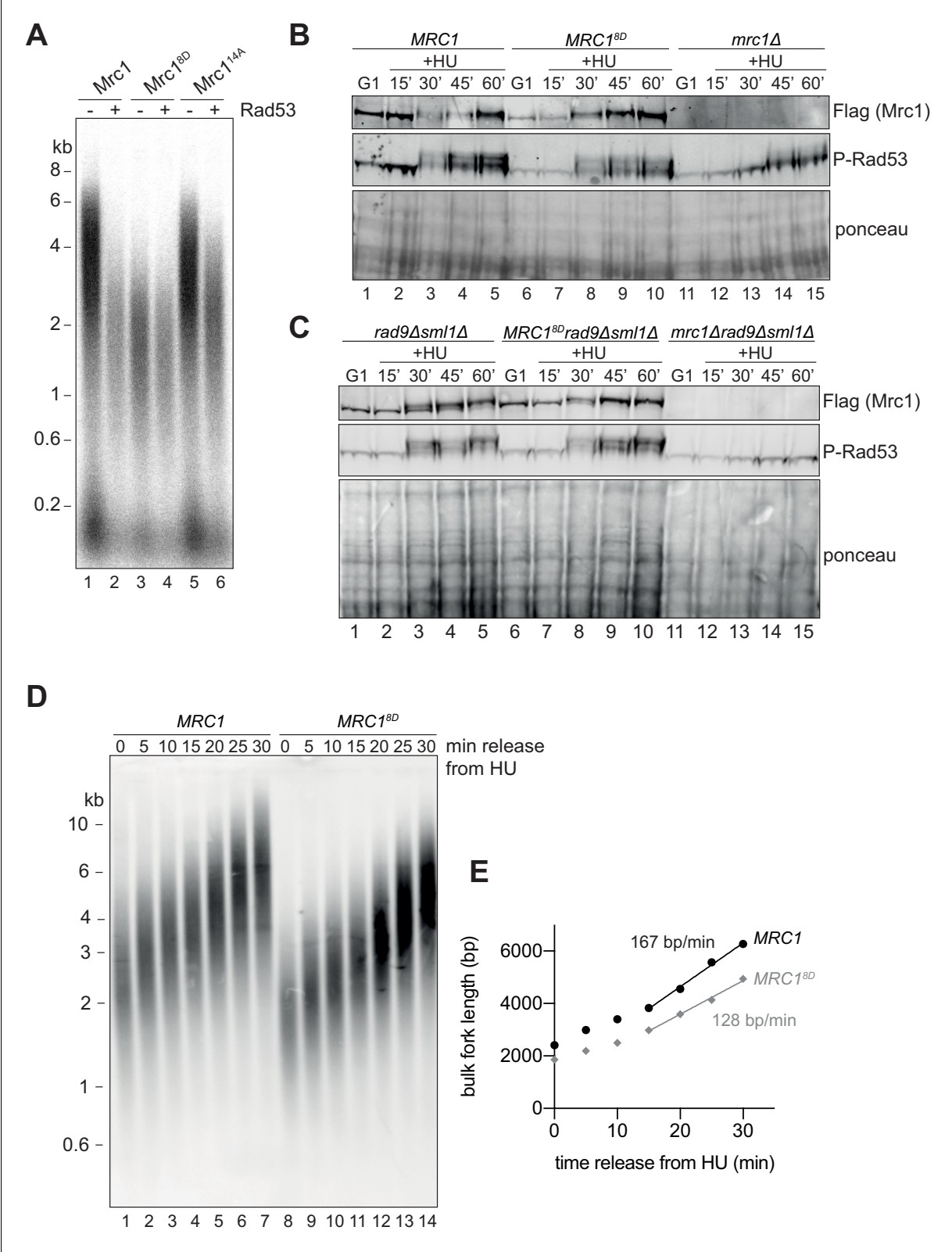

**Figure 6.** Mrc1[8D] slows fork rate in vitro and in vivo. (**A**) Mrc1 and Mrc1[8D] were incubated with Rad53 prior to addition to replication reaction for 7 min. (**B**) Cells harbouring *MRC1-3xFLAG* (yAWM336), *MRC1[8D]-3xFLAG* (yAWM291), or *mrc1Δ* (yAWM217) were synchronised in G1 with α-factor then released into media with or without 200 mM HU for the indicated timepoints. TCA lysates were then analysed by western blot with the indicated antibodies. (**C**) Cells that contained *sml1Δ rad9Δ* in addition to *MRC1-3xFLAG* (yAWM346), *MRC1[8D]-3xFLAG* (yAWM348), or *mrc1Δ* (yJT135) were
*Figure 6 continued on next page*

*Figure 6 continued*

treated as in (B). (D) Cells harbouring *MRC1-3xFLAG* (yAWM373) or *MRC1^{8D}-3xFLAG* (yAWM375) with Gal1~prom~-dmdNK and Gal1~prom~-hENT were synchronised in G1 with α-factor then released into media with BrdU and HU for 1 hr, the released into media with thymidine to chase the BrdU labelling. Following DNA extraction, samples were separated on an alkaline gel, transferred to a nylon membrane, and immunoblotted with anti-BrdU antibody. (E) quantification of samples from (D) (see Materials and methods for detailed method).

The online version of this article includes the following source data for figure 6:

**Source data 1.** Original gel images for *Figure 6*.

without initiating a global checkpoint response. But under more severe replication stress, where more Rad53 is active, Rad53 could phosphorylate Mrc1 at all forks to slow replication globally.

Using a novel assay to measure DNA unwinding activity, we found that unwinding by CMG is not very synchronous, as evidenced by shallow unwinding curves. Moreover, sites as close as 1 kb from the origin were not completely unwound after 50 min. In the presence of Mrc1, unwinding was faster and the curves were steeper, suggesting more synchronous unwinding leading to even the site 2 kb from the origin approaching 100% unwinding by 30 min. In single-molecule experiments, CMG frequently paused and backtracked while unwinding DNA (*Burnham et al., 2019*). The ability to backtrack is thought to release CMG from a non-productive DNA duplex-engaged state in which duplex DNA enters the central CMG channel (*Kose et al., 2020*). Stochastic entry into this state may underlie the asynchronous unwinding curves in our assay, and Mrc1 could stimulate unwinding either by preventing CMG entry into the duplex-engaged state or by promoting its reversal. Regardless, it is interesting to consider that, by inhibiting Mrc1 and allowing CMG to either backtrack or engage the duplex, Rad53 may also contribute to replication fork repair or restart.

A recent structure of CMG bound to M/C/T suggests the C-terminus of Mrc1, containing the majority of its Rad53 phosphorylation sites, may contact Cdc45 and Mcm2 (*Baretić et al., 2020*). Other work has suggested an interaction with the non-catalytic domain of Polε (*Lou et al., 2008*). It would be interesting to understand how these interactions may drive Mrc1's stimulation of CMG activity and how phosphorylation modulates them. The Mrc1^{8D} mutant showed normal checkpoint activation supporting the idea that it remains bound to replication forks. However, we note that Rad53 can target more than these eight sites and full phosphorylation may affect more functions and protein-protein interactions than those disrupted in the Mrc1^{8D} mutant.

Previous work using chromatin immunoprecipitation showed that CMG components moved further away from DNA replication in *mrc1Δ* cells treated with HU, suggesting that Mrc1 has some role in restraining CMG at stalled forks (*Katou et al., 2003*). Superficially, this appears inconsistent with our results: loss of functional Mrc1 might be predicted to lead to slower CMG unwinding, and, therefore, less distance between CMG and DNA replication. However, our results, along with other previous work, suggest an explanation for this. In *mrc1Δ* cells, Rad53 activation is significantly delayed in HU (*Bacal et al., 2018*; *Osborn and Elledge, 2003*). We suggest that during this delay, unphosphorylated Mcm10 continues to drive CMG progression, unwinding further from the stalled DNA synthesis; in wild-type cells on the other hand, rapid Rad53 activation would inactivate both Mrc1 and Mcm10 leading to more rapid slowing of CMG.

*MRC1^{8D}* only weakly suppressed the sensitivity of *rad53Δ* cells to replication stress (*Figure 7*). In these cells, Mcm10 cannot be phosphorylated because Rad53 is absent; consequently, replication fork rate should still be faster than in *RAD53* wild-type cells where both Mrc1 and Mcm10 would be phosphorylated. Thus, it remains unclear how important fork slowing is in Rad53's role in maintaining replication fork stability. Why the cell utilises two targets to regulate fork progression is unclear, and further work is required to understand this.

Studies in mammalian cells have shown that replication forks are slowed down globally in response to replication checkpoint activation (*Mutreja et al., 2018*; *Seiler et al., 2007*). It would be interesting to explore whether Claspin, the mammalian homolog of Mrc1, or Mcm10 might be involved in this process during replication stress.

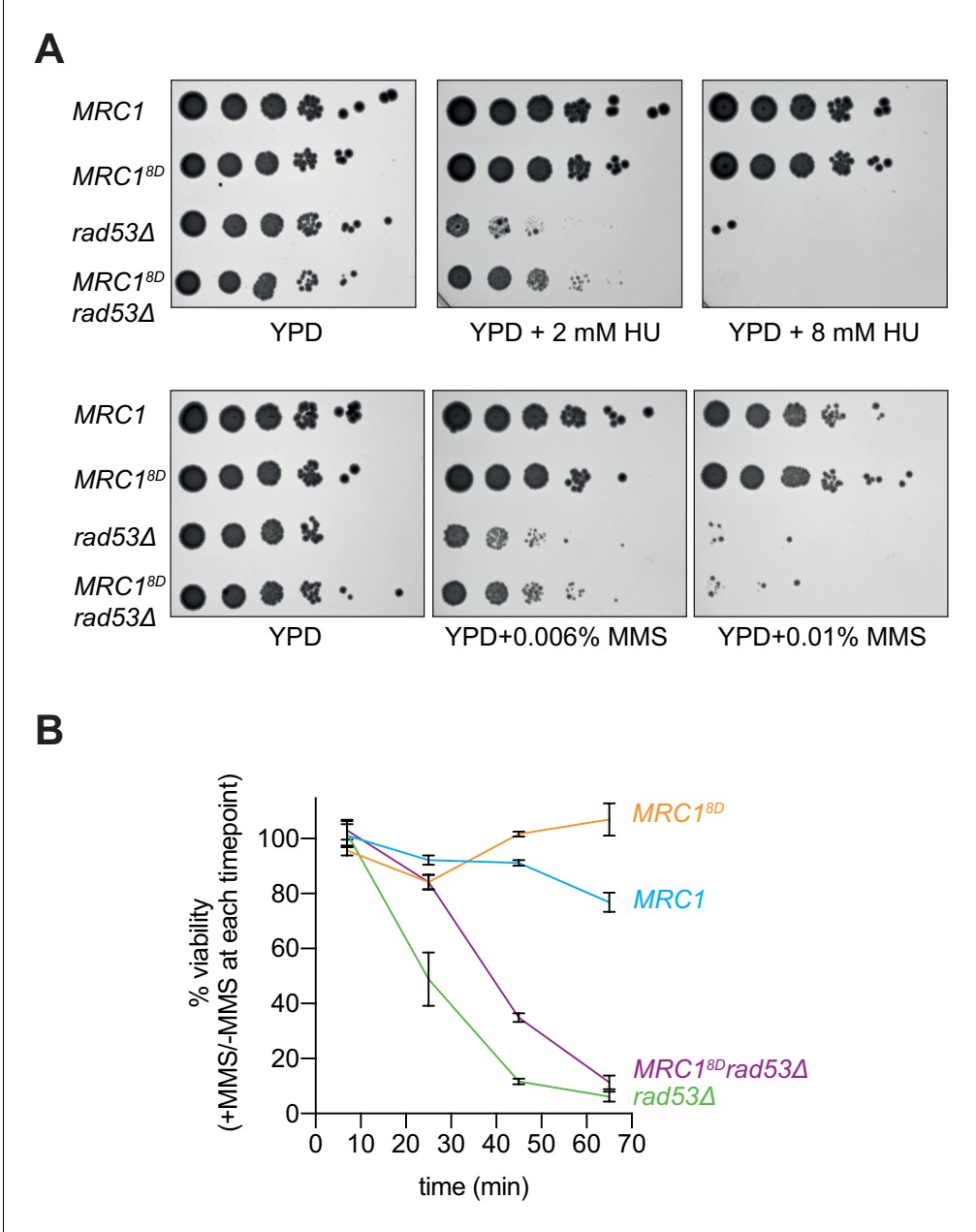

**Figure 7.** Mrc1[8D] partially rescues *rad53Δ* sensitivity to replication stress. (**A**) Cells that contained *sml1Δ* as well as *MRC1-3xFLAG* (yAWM337), *MRC1[8D]-3xFLAG* (yAWM292), *MRC1-3xFLAG* and *rad53Δ* (yAWM338), or *MRC1[8D]-3xFLAG* and *rad53Δ* (yAWM293) were spotted as 1:10 serial dilutions onto YPD plates supplemented with the indicated drugs. (**B**) Cells from (**A**) were arrested in G1 with alpha-factor then released into media with or without 0.02% MMS for the indicated timepoints then plated on YPD plates in triplicate (mean +/- s.e.m.).

The online version of this article includes the following source data and figure supplement(s) for figure 7:

**Source data 1.** Original plate images for *Figure 7A*.

**Source data 2.** Source data for *Figure 7B*.

**Figure supplement 1.** Just after separating tetrad spores, two strains of each of the indicated genotype (with *sml1Δ*) were identified, grown for 6 hr, and then spotted in 1:10 serial dilutions on the indicated plates.

**Figure supplement 1—source data 1.** Original plate images for *Figure 7—figure supplement 1*.

## Materials and methods

### Replication reactions

See *Table 1* for summary of protein purification strategies. All replication proteins were purified as in *Yeeles et al., 2015*; *Yeeles et al., 2017* and Polε^exo- was purified as in *Goswami et al., 2018*. Rad53 was expressed and purified as in *Deegan et al., 2016*; *Gilbert et al., 2001* with an additional gel filtration step at the end in the following buffer: 25 mM HEPES-KOH, 0.02% NP-40-S, 10% glycerol, and 300 mM NaCl.

Replication reactions were essentially performed as in *Yeeles et al., 2017*. Reaction buffer contained: 25 mM HEPES-KOH (pH 7.6), 100 mM potassium glutamate or sodium acetate, 10 mM magnesium acetate, 2 mM DTT, 0.02% NP-40-S, and 5 mM ATP. Experiments were all performed in a thermomixer at 30°C and 1250 rpm. For experiments where proteins were preincubated with Rad53, samples were incubated with 5 mM ATP in reaction buffer for 15–30 min. MCM loading was performed in a master mix of 5 ul per sample with 40 nM ORC, 40 nM Cdc6, and 60 nM MCM-Cdt1 on 4 nM of 10.6 kb plasmid DNA with ARS1 origin. After 20 min, DDK was added to 40–50 nM and further incubated for 10 min. The reaction volume was then doubled with a protein mix and nucleotide mix to give the final concentration of: 20–80 nM Cdc45, 30 nM Dpb11, 20 nM Polε, 20 nM GINS, 15–20 nM CDK, 100 nM RPA, 20 nM Ctf4, 10 nM Topol, 20 nM Csm3/Tof1, 20 nM Mrc1, 20–60 nM Polα, 20–25 nM Sld3/7, 20 nM Mcm10, 20–50 nM Sld2, 200 µM CTP, 200 µM GTP, 200 µM UTP, 80 µM dCTP, 80 µM dGTP, 80 µM dTTP, 80 µM dATP, and 33–50 nM $\alpha^{32}$P-dCTP. In *Figures 4C*, 20 nM PCNA, 20 nM RFC, and 20 nM Polδ, were also added in the last step. Reactions were then stopped by the addition of EDTA, processed over Illustra MicroSpin G-50 columns, separated on alkaline agarose gels, fixed in 5% trichloroacetic acid, dried, exposed to phosphor screens, and scanned using a Typhoon phosphorimager.

**Table 1.** Protein purification strategy.

| Protein | Purification strategy (see *Deegan et al., 2016*; *Yeeles et al., 2015*; *Yeeles et al., 2017* for more details) |
|---|---|
| MCM-Cdt1 | Yeast expression, calmodulin pull-down, EGTA elution, gel filtration |
| Cdc6 | Bacterial expression, glutathione pull-down, precission protease elution, HTP column, dialysis |
| ORC | Yeast expression, calmodulin pull-down, EGTA elution, gel filtration |
| DDK | Yeast expression, calmodulin pull-down, EGTA elution, gel filtration |
| Rad53 | Bacterial expression, Ni-NTA pull-down, imidazole elution, gel filtration |
| Mrc1 | Yeast expression, Flag pull-down, flag peptide elution, MonoQ, dialysis |
| Cdc45 | Yeast expression, Flag pull-down, flag peptide elution, HTP column, dialysis |
| Dpb11 | Yeast expression, Flag pull-down, flag peptide elution, gel filtration |
| Polε and Polε^exo- | Yeast expression, calmodulin pull-down, EGTA elution, heparin column, gel filtration |
| Polε^ΔCAT | Yeast expression, calmodulin pull-down, EGTA elution, MonoQ, gel filtration |
| GINS | Bacterial expression, Ni-NTA pull-down, imidazole elution, MonoQ, gel filtration |
| CDK | Yeast expression, calmodulin pull-down, TEV elution, Ni-NTA column, gel filtration |
| RPA | Yeast expression, calmodulin pull-down, EGTA elution, heparin column, gel filtration |
| Ctf4 | Yeast expression, calmodulin pull-down, EGTA elution, MonoQ, gel filtration |
| Topol | Yeast expression, calmodulin pull-down, EGTA elution, gel filtration |
| Csm3/Tof1 | Yeast expression, calmodulin pull-down, TEV elution, gel filtration |
| Polα | Yeast expression, calmodulin pull-down, EGTA elution, MonoQ, gel filtration |
| Sld3/7 | Yeast expression, IgG Sepharose six pull-down, TEV elution, Ni-NTA column, gel filtration |
| Mcm10 | Bacterial expression, Ni-NTA pull-down, imidazole elution, gel filtration |
| Sld2 | Yeast expression, ammonium sulphate precipitation, Flag pull-down, flag peptide elution, SP column, dialysis |
| RFC | Yeast expression, calmodulin pull-down, EGTA elution, MonoS, gel filtration |
| PCNA | Bacterial expression, ammonium sulphate precipitation, SP column, heparin column, DEAE column, MonoQ, gel filtration |
| Polδ | Yeast expression, calmodulin pull-down, EGTA elution, heparin column, gel filtration |

Four-step replication reactions (*Figures 2A*, *3A and B*) were modified from the three-step reactions above. Following DDK incubation, firing factors were added for 10 min prior to the addition of the remaining proteins and nucleotide mix.

Quantification of signal was performed using FIJI software. Image signal was linearised with 'linearise gel data' plug-in. Distance was then calibrated to base pair length using the base pair ladder with an exponential fit. Each lane was fit with a smooth line using Prism 8, then the leading fork length was defined as the position where the signal was 20% the maximum signal of the lane.

### In vitro kinase assays

For *Figures 1A, C*, *2C*, *3C* and *5C*, Rad53 was incubated at equimolar ratio to target protein with 5 mM ATP for 15 min prior to separation on SDS-PAGE and coomassie stain. For *Figure 5D*, Rad53 was incubated with target protein with 0.2 mM ATP and 0.2 µCi/µl γ$^{32}$P-ATP for 15 min, processed over Illustra MicroSpin G-50 columns, separated on SDS-PAGE, coomassie stained, dried, exposed to phosphor screens, and scanned using a Typhoon phosphorimager.

### Peptide array

Peptide arrays were synthesised on an Intavis ResPep SLi automated synthesiser (Intavis Bioanalytical Instruments AG, Cologne, Germany). The peptides were synthesised using FMOC for temporary α-amino group protection. Protecting groups used were Pbf for arginine, OtBu for glutamic acid and aspartic acid, Trt for asparagine, glutamine, histidine, and cysteine, tBu for serine, threonine and tyrosine, and Boc for lysine and tryptophan. Each amino acid was coupled by activating its carboxylic acid group with DIC in the presence of HOBT. Individual aliquots of amino acids were spotted on to a cellulose membrane which has been derivatised to have 8 to 10 ethylene glycol spacers between the cellulose and an amino group. Synthesis was accomplished by cycles of coupling of amino acids, washing then removal of the temporary α-amino protecting group by piperidine followed by more washing. Once the required number of cycles of coupling and deprotection and washing had been completed, the membranes were treated with a solution of 20 ml containing 95% TFA, 3% TIS, and 2% water for 4 hr. Following this treatment, membranes were washed four times with DCM, four times with ethanol, and twice with water to remove side chain protecting groups and TFA salts and once again with ethanol for easier drying. Just prior to kinase assay, membranes were washed extensively in reaction buffer, then incubated with 80 nM Rad53, 10 µM ATP, and 0.02 µCi/µl γ$^{32}$P-ATP. Membranes were then washed with 1 M NaCl, 1% SDS, and 0.5% phosphoric acid prior to exposure to phosphor screens, and scanned using a Typhoon phosphorimager.

### CMG helicase assay

The CMG helicase assays were performed with a 5 kb template containing an efficient artificial origin (*Coster and Diffley, 2017*) and cassettes of 4 MseI restriction cleavage sites (see *Table 2* and CMG helicase assay template sequence below), which was linearised with ScaI. MCM loading specifically at the origin was done as follows (and visualised with replication reaction in *Figure 4—figure supplement 1*): 5 min incubation of 10 nM DNA, 40 nM Orc, 5 mM ATP, 80 mM NaCl, in replication buffer at 30C and 1250 rpm. Then 40 nM Cdc6 and 60 nM MCM-Cdt1 are added for an additional 10 min followed by passing over a G-50 illustra micro-spin column pre-equilibrated in replication buffer to remove NaCl. ATP and 50 nM DDK are then added for 10 min, then the reaction volume is doubled with a final concentration of: 40–80 nM Cdc45, 30 nM Dpb11, 20 nM Polε$^{exo-}$ (D290A, E292A), 20 nM GINS, 15 nM CDK, 20 nM Csm3/Tof1, 20 nM Mrc1, 20–30 nM Sld3/7, 20 nM Mcm10, 20–50 nM Sld2, and 350 nM RPA. At each time point, 2.5 µl of reaction was added to a tube containing 5 µl replication buffer and 1 µl MseI (NEB) for 3 min, and then the reaction was quenched with the addition of EDTA. Samples were then deproteinated with SDS and proteinase K followed by column clean-up (QIAquick PCR purification kit) according to manufacturer's instructions. The final elution was done with 300 µl 10 mM Tris pH 8. Then qPCR was performed in triplicate using 4 µl sample in 8–9 µl reaction with FastStart Universal SYBR Green Master Mix (Roche) and primers flanking each MseI cassette (*Table 2*).

The raw data was first averaged over the triplicate qPCR reactions, then normalised to the average of the control qPCR reactions of each sample using primers that anneal to the template away from the MseI cleavage sites. The time-course curve for each site (bp from origin) was then fit with a

**Table 2.** Primers used in CMG helicase assay.

| Site (distance from origin) | Primer numbers | Sequences |
|---|---|---|
| 200 bp | AWM107<br>AWM109 | CACTGCACCAAGGTAACACTC<br>GAAGTCAGAGCTGGAGAATCCG |
| 500 bp | AWM111<br>AWM112 | CCCTACTTCAGCGCCATTCG<br>TAACGGAAGCACCGAATCGT |
| 1000 bp | AWM113<br>AWM115 | CTCGTTGTGACGCCAATCAG<br>ACATTGAGCCTACGCATCTGT |
| 1500 bp | AWM78<br>AWM79 | ACTACTGTCACTTCTGAGGGTTC<br>CAGAGGGATGCGTAGTCGTG |
| 2000 bp | AWM116<br>AWM117 | CGGGGGAAGGAACTCTTGC<br>AGGGGTCGTCAAGCAGAGAT |
| control site<br>(not flanking MseI) | AWM84<br>AWM85 | CTCTGCTTGACGACCCCTTG<br>TGTCCGTCCGAGAGCGATA |

spline and normalised to the unwound DNA in the last timepoint of the closest site (200 bp from origin). The unwinding rate was then calculated by integrating the spline and averaging over the last three sites (1000, 1500, and 2000 bp from origin).

## CMG helicase assay template sequence (pAWM36)

AGTACTCAACCAAGTCATTCTGAGAATAGTGTATGCGGCGACCGAGTTGCTCTTGCCCGGCG
TCAATACGGGATAATACCGCGCCACATAGCAGGACTTTAAAAGTGCTCATCATTGGAAAACGTTC
TTCGGGGCGAAAACTCTCAAGGATCTTACCGCTGTTGAGATCCAGTTCGATGTAACCCACTCG
TGCACCCAACTGATCTTCAGCATCTTTTACTTTCACCAGCGTTTCTGGGTGAGCAAAAACAG-
GAAGGCAAAATGCCGCAAAAAAGGGAATAAGGGCGACACGGAAATGTTGAATACTCATACTC
TTCCTTTTTCAATATTATTGAAGCATTTATCAGGGTTATTGTCTCATGAGCGGATACATATTTGAATG
TATTTAGGGGAATAAACAAATAGGGGTTCCGCGCACATTTCCCCGAAAAGTGCCACCTGACGTC
TAAGAAACCATTATTATCATGACATTGGCCTATAAAAATAGGCGTATCACGAGGCCCTTTCGTC
TCGCGCGTTTCGGTGATGACGGTGAAAACCTCTGACACATGCAGCTCCCGGAGACGGTCACAGC
TTGTCTGTAAGCGGATGCCGGGAGCAGACAAGCCCGTCAGGGCGCGTCAGCGGGTG
TTGGCGGGTGTCGGGGCTGGCTTAACTATGCGGCATCAGAGCAGATTGTACTGAGAGTGCACCA
TATGCGGTGTGAAATACCGCACAGATGCGTAAGGAGAAAATACCGCATCAGGCGCCATTCGCCA
TTCAGGCTGCGCAACTGTTGGGAAGGGCGATCGGTGCGGGCCTCTTCGCTATTACGCCAGC
TGGCGAAAGGGGGATGTGCTGCAAGGCGATTAAGTTGGGTAACGCCAGGGTTTTCCCAGTCAC-
GACGTTGTAAAACGACGGCCAGTGAATTCCTCGATTTTTTTATGTTTAGTTTCGCGGACGACGG
TTTCGAGGTGGCGGTCTGGACCACGCCGGAGAGCGTCGAAGCGGAGGCGGTGTTCGCCGAGA
TCGGCTCGCGCAAAGCCGAGTTGAGCGAACTAAACATAAAAATACAGCATCAGATGGTAGGCC
TCCTGGCGCCGCACCGGCCTCAGCATCCGGTACCTCAGCTGGCCACATCACTGTCTTTCTTA
TGACGGTACTACCGGTGTTCACTGCACCAAGGTAACACTCATTAAATTAAGGTTAAATTAATC
TACACAATTCTCTTTTGCTATTGGTACCGGATTCTCCAGCTCTGACTTCAGCGTCTCTGAAGGAATC
TTTGCAGGTGCTTACGCTTACTACCTAAACTACAATGGTGTTGTCGCTACTAGTGCCGCTTC
TTCAACCACTGGATCTGGTCCTAGGGCTTCGGTCCGCCCCTACTTCAGCGCCATTCGCCA
TTCAGGCTGCGCAACTGTTGGGAAGGGCGATCGGTGCGGGCCTCTTCGCTATTACGCCAGC
TGGCGAAAGGGGGATGTGCTGCAATTCGGTTAAGTTAAGTTAAGGTTAAAGAAGCTAACGC-
CAGGGTTTTCCCAGTCACGATTCGGTGCTTCCGTTACCGGTTCAACTGCTTCCACTTCATGGGC
TACTTTTTGGACCGGAACGGCTGGTACTATCGGCCTGGTATCATCCTTTACCGAAGCAACATCTG
TTTACACTACAACACTAGACCAAGCACAGTCGTAGTTTCTTGTTCAGAGATGACTCCAATGG
TAACGTCTATACCATTACCACAATCATTAAGTTAAATTAAGTTAATCAAACCGTTCCATGCTCA
TCCACTACCGCCACTATTACTTCTTGTGATGAAACTGGATGTCACGTTAGTACATCAACCGGTGC
TGTTGTAACTGAAACCGTTTCTTCCAAGGCATACACAACTGCCAAAGTAACTCGTTGTGACGCCAA
TCAGCTTGTCTGTAAGCGGATGCCGGGAGCAGACAAGCCCGTCAGGGCGCGTCAGCGGGTG
TTGGCGGGTGTCGGGGCATGGTTAAAGTTAACTTAAATTAAGGTACGCGGCATCAGAGCAGA
TTGTACTGAGAGTGCACCATATGCGGTGTGAAATACCGCACAGATGCGTAGGCTCAATGTAAC
TTAGCCACTGTCAATTGGGAATGTTCCAGGGATTCATGGACAACAACTGCAACTGGAGTATCA

TACACCACTGTCACCGTAACCCACTGTGACGACAATGGCTGTAACACCAAGACTAAGCTCC
TGAAGCTACCACCACAACTATCGCCCACCAGGACCACCGTCACCTTTAGTGATGACAATGAAGG
TAAGACCTTGGGTGAGTCTGGTCCAGCGGAGGGCCACTACTGTTTCTCCAAAGACATACAC-
CACCGCTACTGTTACTCAGGGGGATAAAAATGCCTGCCTCACCAAGACTGTCACTTCTGAATG
TCCTGAAGAAACTTCAGCAACTACTACTGTCACTTCTGAGGGTTCTAAAGCAACCTCATTGAG
TCGACGCGGGGGCGACGATTAACCTTAACGTTAAGTTAAGCTAGCACGACTACGCATCCCTC
TGACTACTTCTCGGGGTGGGACTATACTGGTACCGATACGGGCTGTGATGACAACGATGTG
TAGAACTGGGACAATCAGATCTGAGGCCCCTGAAGCCACAACGGGTACTGTTTCTAACAACAGA
TACAACATGGAGGGCCAACATTGTCACAATAGAAGCTCCGCCAGAAACAGTAGAAACTTCA-
GAAACCAGTGCTGCCCCTAAGGACATACACTACTGCCACTGGTTACTCAATGGTTTAGAGGGTGG
TTGCCACGTCAAGATAATCACCTCTAAAATACCTGAAGCTACTTCAACCGTCACGGGTGCTTC
TCCAAAACGGCCTTACATAGCCGGATACAGTGACTTTGACAGGTTTGCGGGGCACAGCAATGAC
TTGCATAGCTGCGTGCGGGGGAAGGAACTCTTGCGTCTCACTGCCCGCTTTCCAG
TCGGGAAACCTGTCGTGCCATTATGGTTAACTTAAGTTAATTTAAGCTATCGGCCAACGCGCGGG-
GAGAGGCGGTTTGCGTATTGGGCGCTCTTCCGCTTCCTCGCTCAGTGAGTATCTCTGCTTGAC-
GACCCCTTGGCGCAGAGGTGCTGGCCGCGTGCTAAGTTGAAGCGGCTGCACTGCTGCAAGG
TCCGTCACGGAGGCGTCGGACCGGCAGGAGCACTAGCCCATCGACCCGTACGGGAACACTCTA
TATCGCTCTCGGACGGACATTCTGGATCCTCTAGAGTCGACCTGCAGGCATGCAAGCTTGGCG
TAATCATGGTCATAGCTGTTTCCTGTGTGAAATTGTTATCCGCTCACAATTCCACACAACATAC-
GAGCCGGAAGCATAAAGTGTAAAGCCTGGGGTGCCTAATGAGTGAGCTAACTCACATTAA
TTGCGTTGCGCTCACTGCCCGCTTTCCAGTCGGGAAACCTGTCGTGCCAGCTGCATTAATGAA
TCGGCCAACGCGCGGGGAGAGGCGGTTTGCGTATTGGGCGCTCTTCCGCTTCCTCGCTCAC
TGACTCGCTGCGCTCGGTCGTTCGGCTGCGGCGAGCGGTATCAGCTCACTCAAAGGCGGTAA
TACGGTTATCCACAGAATCAGGGGATAACGCAGGAAAGAACATGTGAGCAAAAGGCCAG-
CAAAAGGCCAGGAACCGTAAAAAGGCCGCGTTGCTGGCGTTTTTCCATAGGCTCCGCCCCCC
TGACGAGCATCACAAAAATCGACGCTCAAGTCAGAGGTGGCGAAACCCGACAGGACTA
TAAAGATACCAGGCGTTTCCCCCTGGAAGCTCCCTCGTGCGCTCTCCTGTTCCGACCCTGCCGC
TTACCGGATACCTGTCCGCCTTTCTCCCTTCGGGAAGCGTGGCGCTTTCTCATAGCTCACGCTG
TAGGTATCTCAGTTCGGTGTAGGTCGTTCGCTCCAAGCTGGGCTGTGTGCACGAACCCCCCG
TTCAGCCCGACCGCTGCGCCTTATCCGGTAACTATCGTCTTGAGTCCAACCCGGTAAGACACGAC
TTATCGCCACTGGCAGCAGCCACTGGTAACAGGATTAGCAGAGCGAGGTATGTAGGCGGTGC
TACAGAGTTCTTGAAGTGGTGGCCTAACTACGGCTACACTAGAAGAACAGTATTTGGTATC
TGCGCTCTGCTGAAGCCAGTTACCTTCGGAAAAAGAGTTGGTAGCTCTTGATCCGGCAAA-
CAAACCACCGCTGGTAGCGGTGGTTTTTTTGTTTGCAAGCAGCAGATTACGCGCA-
GAAAAAAAGGATCTCAAGAAGATCCTTTGATCTTTTCTACGGGGTCTGACGCTCAGTGGAAC-
GAAAACTCACGTTAAGGGATTTTGGTCATGAGATTATCAAAAAGGATCTTCACCTAGATCC
TTTTAAATTGGAAATGAAGTTTTAAATCAATCTAAAGTATATATGAGTAAACTTGGTCTGACAG
TTACCAATGCTTAATCAGTGAGGCACCTATCTCAGCGATCTGTCTATTTCGTTCATCCATAGTTGCC
TGACTCCCCGTCGTGTAGATAACTACGATACGGGAGGGCTTACCATCTGGCCCCAGTGCTGCAA
TGATACCGCGAGACCCACGCTCACCGGCTCCAGATTTATCAGCAATAAACCAGCCAGCCG-
GAAGGGCCGAGCGCAGAAGTGGTCCTGCAACTTTATCCGCCTCCATCCAGTCTATTAATTG
TTGCCGGGAAGCTAGAGTAAGTAGTTCGCCAGTTAATAGTTTGCGCAACGTTGTTGCCATTGC
TACAGGCATCGTGGTGTCACGCTCGTCGTTTGGTATGGCTTCATTCAGCTCCGGTTCCCAACGA
TCAAGGCGAGTTACATGATCCCCCATGTTGTGCAAAAAAGCGGTTAGCTCCTTCGGTCCTCCGA
TCGTTGTCAGAAGTAAGTTGGCCGCAGTGTTATCACTCATGGTTATGGCAGCACTGCATAATTCTC
TTACTGTCATGCCATCCGTAAGATGCTTTTCTGTGACTGGTG.

## Western blot

Log-phase yeast cultures in YPD were diluted to $OD_{600}$ 0.5 and arrested with 20 µg/ml of alpha-factor for 2–3 hr at 25℃. Cells were washed two times with YPD and then resuspended in YPD + 200 mM HU. Cells were then harvested at the indicated times, and protein was extracted with 10% trichloroacetic acid. Extracts were then processed by 3–8% Tris-acetate SDS-PAGE, transferred to nitrocellulose, and immunoblotted with anti-Flag (M2 mouse monoclonal) and anti-Rad53 (Abcam, ab104232, rabbit polyclonal) antibodies.

**Table 3.** Yeast strains generated in this study.

| Strain | Genotype (all in W303 background) |
|---|---|
| yVP8 | MATa<br>bar1::hyg$^R$<br>pep4::kan$^R$<br>his3:pRS303-Gal1$_{prom}$-3xFLAG-DPB11:HIS3 |
| yVP7 | MATa<br>bar1::hyg$^R$<br>pep4::kan$^R$<br>his3:pRS303-Gal1$_{prom}$-MRC1$^{17AQ}$-3xFLAG:HIS3 |
| yAWM106 | MATa<br>bar1::hyg$^R$<br>pep4::kan$^R$<br>his3:pRS303-Gal1$_{prom}$-MRC1$^{1-875}$-3xFLAG:HIS3 |
| yAWM107 | MATa<br>bar1::hyg$^R$<br>pep4::kan$^R$<br>his3:pRS303-Gal1$_{prom}$-MRC1$^{8D}$-3xFLAG:HIS3 |
| yAWM105 | MATa<br>bar1::hyg$^R$<br>pep4::kan$^R$<br>his3:pRS303-Gal1$_{prom}$-MRC1$^{14A}$-3xFLAG:HIS3 |
| yAWM115 | MATa<br>bar1::hyg$^R$<br>pep4::kan$^R$<br>his3:pRS303-Gal1$_{prom}$-MRC1$^{19A}$-3xFLAG:HIS3:NAT |
| yAWM108 | MATa<br>bar1::hyg$^R$<br>pep4::kan$^R$<br>his3:pRS303-Gal1$_{prom}$-MRC1$^{41A}$-3xFLAG:HIS3 |
| yAWM343 | MATa/α<br>Mrc1$^{8D}$-3xFLAG:nat$^R$/MRC1<br>sml1::kan$^R$/SML1<br>rad9::LEU2/RAD9 |
| yAWM373 | MATa<br>MRC1-3xFLAG:nat$^R$ leu2: Gal1$_{prom}$-hENT:LEU2<br>trp1: Gal1$_{prom}$-dmdNK:TRP1 |
| yAWM375 | MATa<br>MRC1$^{8D}$-3xFLAG:nat$^R$ leu2: Gal1$_{prom}$-hENT:LEU2<br>trp1: Gal1$_{prom}$-dmdNK:TRP1 |
| yAWM337 | MATa<br>MRC1-3xFLAG:nat$^R$ sml1::kan$^R$ |
| yAWM292 | MATa<br>MRC1$^{8D}$-3xFLAG:nat$^R$ sml1::kan$^R$ |
| yAWM338 | MATa<br>MRC1-3xFLAG:nat$^R$ rad53::LEU2<br>sml1::kan$^R$ |
| yAWM293 | MATa<br>MRC1$^{8D}$-3xFLAG:nat$^R$ rad53::LEU2<br>sml1::kan$^R$ |
| yAWM336 | MATa<br>MRC1-3xFLAG:nat$^R$ |
| yAWM233 | MATa<br>MRC1$^{19A}$:nat$^R$ |
| yAWM291 | MATa<br>MRC1$^{8D}$-3xFLAG:nat$^R$ |
| yAWM217 | MATa<br>mrc1::kan$^R$ |
| yAWM346 | MATa<br>MRC1-3xFLAG:nat$^R$ rad9::LEU2<br>sml1::kan$^R$ |

*Table 3 continued on next page*

**Table 4.** DNA plasmids generated in this study.

| Plasmid | Description *see details in the construction notes |
|---|---|
| pAWM7 | pET21b-RAD53$^{K227A,D339A}$-6xHis |
| pAWM10 | pET21b-MRC1$^{1-283}$-6xHis |
| pAWM11 | pET21b-MRC1$^{110-430}$-6xHis |
| pAWM12 | pET21b-MRC1$^{355-670}$-6xHis |
| pAWM13 | pET21b-MRC1$^{555-900}$-6xHis |
| pAWM14 | pET21b-MRC1$^{869-1096}$-6xHis |
| ~~pAWM16~~ | ~~pRS303-GAL1$_{prom}$-MRC1-3xFLAG, GAL4~~ |
| pVP14 | pRS303-GAL1$_{prom}$-MRC1$^{17AQ}$-3xFLAG, GAL4 |
| pAWM35 | pRS303-GAL1$_{prom}$-MRC1$^{1-875}$-3xFLAG, GAL4 |
| pAWM15 | pRS303-GAL1$_{prom}$-MRC1$^{14A}$-3xFLAG, GAL4 |
| pAWM25 | pRS40N- MRC1$^{19A}$-3xFLAG (N-terminal truncation for integration) |
| pAWM18 | pRS303-GAL1$_{prom}$-MRC1$^{41A}$-3xFLAG, GAL4 |
| pAWM17 | pRS303-GAL1$_{prom}$-MRC1$^{8D}$-3xFLAG, GAL4 |
| pAWM48 | pRS40N-MRC1-3xFLAG (N-terminal truncation for integration) |
| pAWM47 | pRS40N-MRC1$^{8D}$-3xFLAG (N-terminal truncation for integration) |
| pAWM36 | pBS-based template for CMG helicase assay |
| pAWM37 | pBS-based template for CMG helicase assay (no origin) |

## Replication intermediate detection in yeast with BrdU-labelling

Log-phase yeast cultures in YP-raffinose were centrifuged and resuspended in YP-galactose to OD$_{600}$ 0.3 and arrested with 20 µg/ml of α-factor for 3 hr at 25°C. Cells were washed two times with YPG and then resuspended in YPG + 200 mM HU + 200 µg/ml BrdU. After 1 hr, cells were washed two times with YPG and then resuspended in YPG + 2 mM thymidine. Cells were harvested at the indicated times, and DNA was extracted with phenol-chloroform. DNA was run on a 0.8% alkaline agarose gel, then alkaline transferred to a positively charged nylon membrane with the VacuGene XL Vacuum Blotting System (Amersham), and immunoblotted with anti-BrdU antibody (BD, 347580, mouse monoclonal).

## Yeast and plasmid strain construction

Yeast strains are listed in *Table 3*. Plasmids are listed in *Table 4*. RAD53$^{K227A,K339A}$ was constructed by PCR with mutated oligos on pET21b-RAD53 (*Gilbert et al., 2001*). Mrc1 fragments were made by PCR from genomic DNA and cloned into pET21b vector with NheI and XhoI. pAWM16, pVP14, pAWM35, pAWM15, pAWM17, and pAWM18 were derived from the original plasmid yJY17 with the codon-optimised sequence of *MRC1* (*Yeeles et al., 2017*) and modified with Gibson assembly methods, and then transformed into the *his3* locus by cleaving with NheI. pAWM25 contained the C-terminal portion of MRC1$^{19A}$ (starting from base pair 1395) and the 3xFLAG tag between the BamHI and NotI sites of pRS40N and then was cut with Blp1 to modify the already integrated codon-optimised MRC1 at the *his3* locus. pVP14 (MRC1$^{17AQ}$) contains all S/T residues followed by Q sites mutated to A as in *Osborn and Elledge, 2003*. pAWM15 (MRC1$^{14A}$) contains the following mutations: S911A, S918A, S920A, T952A, S957A, T996A, T997A, S1006A, S1033A, T1036A, S1039A, S1040A, S1043A, T1045A. pAWM25 (MRC1$^{19A}$) contains the following mutations: T882A, S911A, S918A, S920A, S924A, T932A, T952A, S957A, S958A, T996A, S997A, S1006A, S1010A, S1033A, T1036A, S1039A, S1040A, S1043A, T1045A. pAWM18 (Mrc1$^{41A}$) contains the following mutations: T882A, S911A, S918A, S920A, S924A, T932A, S937A, T952A, S957A, S958A, S961A, T963A, S965A, T967A, S969A, T970A, T971A, S972A, T974A, T977A, T996A, S997A, S1006A, S1010A, S1013A, T1015A, T1027A, S1033A, T1036A, S1039A, S1040A, S1043A, T1045A, T1050A, T1060A, T1063A, T1079A, T1081A, S1083A, S1089A, S1093A. pAWM17 (Mrc1$^{8D}$) contains the following mutations: S911D, S918D, S920D, T952D, S957D, T996D, S997D, S1006D. pAWM47 and

pAWM48 contained base pairs 1424–3288 of MRC1 between XhoI and BamHI and 127 bp of the 3′ UTR of MRC1 between BamHI and NotI. The 3xFLAG tag was added with flanking BamHI sites, and the plasmids were integrated into yeast after cutting with XbaI. The template used in the CMG helicase assay was made by modifying a plasmid with a pBlueScript vector and contained a synthetic origin with either two ORC binding sites 70 bp apart (pAWM36) or no ORC binding sites (pAWM37) (*Coster and Diffley, 2017*) and a synthetic fragment from GeneArt (Thermofisher) that contained the tandem MseI sites. See full pAWM36 plasmid sequence below. The hENT and dmdNK constructs were crossed from strains from *Segurado and Diffley, 2008*.

## Acknowledgements

We thank A Alidoust, N Patel, and D Patel in the Structural Biology Laboratory for yeast protein expression. We thank D Joshi in the Peptide Chemistry Laboratory for generating the Mrc1 peptide arrays and D Frith in the Proteomics Laboratory for processing mass spectrometry samples. We thank V Posse for providing Mrc1$^{17AQ}$ protein. We thank V Posse, A Bertolin, B Canal, and all members of the Diffley group for stimulating discussions and advice. This work was supported by the Francis Crick Institute, which receives its core funding from Cancer Research UK (FC001066), the UK Medical Research Council (FC001066), and the Wellcome Trust (FC001066). This work was also funded by an EMBO Long-Term Fellowship (ALTF 1259–2016 to AWM), Wellcome Trust Senior Investigator Awards (106252/Z/14/Z and 219527/Z/19/Z to JFXD), and a European Research Council Advanced Grant (669424-CHROMOREP to JFXD). For the purpose of Open Access, the author has applied a CC BY public copyright licence to any Author Accepted Manuscript version arising from this submission.

## Additional information

### Funding

| Funder | Grant reference number | Author |
|---|---|---|
| Wellcome Trust | 106252/Z/14/Z | John FX Diffley |
| Wellcome Trust | 219527/Z/19/Z | John FX Diffley |
| Cancer Research UK | FC001066 | John FX Diffley |
| Medical Research Council | FC001066 | John FX Diffley |
| Wellcome Trust | FC001066 | John FX Diffley |
| EMBO | ALTF 1259–2016 | Allison McClure |
| European Research Council | 669424-CHROMOREP | John FX Diffley |

The funders had no role in study design, data collection and interpretation, or the decision to submit the work for publication.

### Author contributions

Allison W McClure, Conceptualization, Formal analysis, Validation, Investigation, Visualization, Methodology, Writing - original draft, Writing - review and editing; John FX Diffley, Conceptualization, Supervision, Funding acquisition, Writing - original draft, Project administration, Writing - review and editing

### Author ORCIDs

Allison W McClure  https://orcid.org/0000-0002-5321-2483
John FX Diffley  https://orcid.org/0000-0001-5184-7680

### Decision letter and Author response

Decision letter https://doi.org/10.7554/eLife.69726.sa1
Author response https://doi.org/10.7554/eLife.69726.sa2

## Additional files

**Supplementary files**
• Transparent reporting form

**Data availability**
Source data files have been provided for all figures.

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
