## [Decision Letter]

**Acceptance summary:**

This paper addresses an important role for the DNA replication checkpoint kinase Rad53 in the yeast *S. cerevisiae* (Chk2 in mammals) in blocking both initiation of DNA replication and replication fork progression via inhibition of Mrc1 activation of the replicative DNA helicase. The paper elegantly utilizes the power of biochemical reconstitution of complete DNA replication with purified proteins and its regulation by checkpoint kinases.

**Decision letter after peer review:**

Thank you for submitting your article "Rad53 checkpoint kinase regulation of DNA replication fork rate via Mrc1 phosphorylation" for consideration by *eLife*. Your article has been reviewed by 3 peer reviewers, including Bruce Stillman as the Reviewing Editor and Reviewer #1, and the evaluation has been overseen by Kevin Struhl as the Senior Editor. The following individual involved in review of your submission has agreed to reveal their identity: Oscar M Aparicio (Reviewer #3).

We all felt that this is a very strong paper that confirms the effect of Rad53 kinase on Sld3 and Dbf4 activities in initiation of DNA replication, but goes further and identifies Mrc1 and Mcm10 as substrates for the kinase in regulating replication fork progression and CMG activity. The results are clear and support the conclusions.

Essential revisions:

The reviewers made a number of suggestions, but only a subset of these are listed below and we ask that you consider adding the requested data.

(1) Page 10 and Figure 5E. The observation that Mrc119A can still be inhibited by Rad53 kinase, albeit less than the wild type Mrc1 protein, might suggest (1) that there are additional phosphorylation sites (as suggested by the authors) or (2) that Rad53 binds directly to Mrc1 and this interaction may partially block its stimulation of CMG. Can the authors test if the Rad53 binds to purified Mrc1? The reason for asking this question is that it is possible that both Mrc1 and Rad53, which are both known to be located at active DNA replication forks, create a solid-state system, feedback control at the replication fork for Mec1 activating Rad53 kinase and Rad53 phosphorylating Mrc1.

(2) How selective is the effect of Rad53 on the Mrc1-dependent replisome slowdown in the biochemical assays? It would be useful to exclude the possibility that any active kinase could phosphorylate Mrc1 and cause the same phenotype.

(3) In addition to the viability assays shown in Figure 6, it would be useful if the authors could provide an assay that monitors replication fork progression in vivo to ensure that the Mrc1-8D protein restrains fork progression even in the absence of Rad53.

*Reviewer #1 (Recommendations for the authors):*

1. Page 10 and Figure 5E. The observation that Mrc119A can still be inhibited by Rad53 kinase, albeit less than the wild type Mrc1 protein, might suggest (1) that there are additional phosphorylation sites (as suggested by the authors) or (2) that Rad53 binds directly to Mrc1 and this interaction may partially block its stimulation of CMG. Can the authors test if the Rad53 binds to purified Mrc1? The reason for asking this question is that it is possible that both Mrc1 and Rad53, which are both known to be located at active DNA replication forks, create a solid-state system, feedback control at the replication fork for Mec1 activating Rad53 kinase and Rad53 phosphorylating Mrc1.

*Reviewer #2 (Recommendations for the authors):*

The key point that needs to be addressed, in my opinion, is whether the phenomena identified in the biochemical studies are reflective of the in vivo situation. I offer below a few comments that should help addressing this issue.

(1) How selective is the effect of Rad53 on the Mrc1-dependent replisome slowdown in the biochemical assays? I ask this since recombinant Rad53 is highly active and it would be useful to exclude the possibility that any active kinase could phosphorylate Mrc1. Alternatively, does the effect of Rad53 on in vitro DNA replication dependent on its two FHA domains?

(2) While I do have sympathy with the authors for their struggle in identifying the key Rad53 target sites on Mrc1, I feel they should nevertheless provide evidence that one or more of the sites mutated in the Mrc1-8D variant are indeed phosphorylated in cells, in a Rad53-dependent manner, in response to DNA replication stress.

(3) In addition to the viability assays shown in Figure 6, it would be useful if the authors could provide an assay that monitors replication fork progression to ensure that the Mrc1-8D protein restrains fork progression even in the absence of Rad53.

*Reviewer #3 (Recommendations for the authors):*

I would really appreciate a figure showing the locations of the 8/14/19 mutations. I feel these should be added to the diagram in S3B.

I suggest the authors construct and test an 8A version. If this allele exhibits more limited effects than 14A, it will strengthen the inference that the 8D changes are not just breaking the protein.

I think a bit more in vivo analysis of these mutants would be useful and appreciated, especially by experts such as bulk analysis of DNA content during S phase progression. One would predict that the 8D allele would cause slow replication even without drugs. However, it might have little effect given the apparently intact checkpoint signaling and presumably intact origin firing control.

One set of easy experiments that seems to be missing is more analysis of the 14A and 19A alleles in vivo. These alleles might be predicted to exhibit sensitivity to HU and MMS. Should effects be noted, it will probably be important to examine Rad53 activation in these mutants similar to analyses in 6B and C to rule out possible effects on transducing the signal to activate Rad53 as opposed to defects in receiving effector signals from activated Rad53. It is possible that the 14A and 19A mutations will have no sensitivities due to incomplete penetrance or redundant mechanisms, but worth a try.

---

## [Author Response]

Essential revisions:The reviewers made a number of suggestions, but only a subset of these are listed below and we ask that you consider adding the requested data.(1) Page 10 and Figure 5E. The observation that Mrc119A can still be inhibited by Rad53 kinase, albeit less than the wild type Mrc1 protein, might suggest (1) that there are additional phosphorylation sites (as suggested by the authors) or (2) that Rad53 binds directly to Mrc1 and this interaction may partially block its stimulation of CMG. Can the authors test if the Rad53 binds to purified Mrc1? The reason for asking this question is that it is possible that both Mrc1 and Rad53, which are both known to be located at active DNA replication forks, create a solid-state system, feedback control at the replication fork for Mec1 activating Rad53 kinase and Rad53 phosphorylating Mrc1.

The reviewer correctly points out that Rad53 might inhibit Mrc1 by binding and not just by phosphorylating Mrc1, as has been suggested for another Rad53 target, DDK (Abd Wahab and Remus, 2020). We think this is unlikely because pre-incubation of Mrc1 with Rad53^KD^ (Figure 2D) does not slow replication. However, we have now included new data (new Figure 2 —figure supplement 1A) showing that neither wild type Rad53 nor Rad53^KD^ co-immunoprecipitate appreciably with Mrc1. Under the same buffer conditions — identical to the buffer conditions in the replication reactions — Csm3/Tof1 co-immunoprecipitate nearly stoichiometrically. We think that, together, these results indicate that Rad53 inhibition of Mrc1 is likely via phosphorylation and not binding. We have added this point in the section “Rad53 inhibition of replication elongation via Mrc1 and Mcm10” on page 6.

(2) How selective is the effect of Rad53 on the Mrc1-dependent replisome slowdown in the biochemical assays? It would be useful to exclude the possibility that any active kinase could phosphorylate Mrc1 and cause the same phenotype.

It is, of course, very difficult to exclude the possibility that some other kinase(s) may also inhibit Rad53. Nonetheless, we think this is an interesting point, especially given the fact that DDK has been implicated in Mrc1 regulation in fission yeast. We, therefore, tested whether the two main replicative kinases, DDK and CDK, can phosphorylate and inhibit Mrc1. In a new Figure 2 —figure supplement 1B we show that neither DDK nor CDK were able to inhibit Mrc1 in replication elongation. We have added this point to section “Rad53 inhibition of replication elongation via Mrc1 and Mcm10” on page 6.

(3) In addition to the viability assays shown in Figure 6, it would be useful if the authors could provide an assay that monitors replication fork progression in vivo to ensure that the Mrc1-8D protein restrains fork progression even in the absence of Rad53.

This is a good point, but not a trivial request. We include new data that we believe at least partially addresses this issue. We previously showed that we could do ‘pulse-chase’ experiments in vivo using yeast strains that express the *Drosophila melanogaster* deoxyribonucleoside kinase (dmdNK) and the human equilibrative nucleoside transporter (hENT1) which together allow efficient BrdU incorporation and rapid chasing of BrdU from nucleotide pools (Segurado and Diffley, 2008; Vernis et al. 2003). In these experiments, cells are released from α factor into HU in medium containing BrdU. This results in a smear of short, labelled replication intermediates from early firing origins on alkaline agarose gels detected by ‘western’ blotting with anti-BrdU antibody. We can then estimate fork rates from the extension of this labelled smear to larger sizes after HU is washed out and BrdU is replaced with thymidine. We have done this in *RAD53^+^* cells carrying wild-type *MRC1* or *MRC1^8D^.* The replication intermediates of *MRC1^8D^* cells were shorter and were extended at a slower rate consistent with the idea that *MRC1^8D^* promotes slower replication forks in vivo (new Figure 6D,E). We have added this to the “Mrc1^8D^ slows fork rate in vitro and in vivo and partially rescues *rad53* mutant sensitivity” section on page 11. Unfortunately, this approach requires synchronisation with HU, and therefore cannot be used with *rad53* mutants because of the extensive fork collapse and continued origin firing in HU (see, for example, Figure 5B in Segurado and Diffley, 2008). Rad53 becomes rapidly dephosphorylated after release from HU, so the latter part of the curve in the new Figure 6E represents forks moving in the absence of active Rad53. Although we recognise this does not exactly address the reviewer’s question, it is probably the best we can do at the moment, and it shows that the 8D mutant does indeed have slower forks in vivo.

Reviewer #1 (Recommendations for the authors):1. Page 10 and Figure 5E. The observation that Mrc119A can still be inhibited by Rad53 kinase, albeit less than the wild type Mrc1 protein, might suggest (1) that there are additional phosphorylation sites (as suggested by the authors) or (2) that Rad53 binds directly to Mrc1 and this interaction may partially block its stimulation of CMG. Can the authors test if the Rad53 binds to purified Mrc1? The reason for asking this question is that it is possible that both Mrc1 and Rad53, which are both known to be located at active DNA replication forks, create a solid-state system, feedback control at the replication fork for Mec1 activating Rad53 kinase and Rad53 phosphorylating Mrc1.

Addressed in Essential revisions.

Reviewer #2 (Recommendations for the authors):The key point that needs to be addressed, in my opinion, is whether the phenomena identified in the biochemical studies are reflective of the in vivo situation. I offer below a few comments that should help addressing this issue.(1) How selective is the effect of Rad53 on the Mrc1-dependent replisome slowdown in the biochemical assays? I ask this since recombinant Rad53 is highly active and it would be useful to exclude the possibility that any active kinase could phosphorylate Mrc1. Alternatively, does the effect of Rad53 on in vitro DNA replication dependent on its two FHA domains?

Addressed in Essential revisions.

(2) While I do have sympathy with the authors for their struggle in identifying the key Rad53 target sites on Mrc1, I feel they should nevertheless provide evidence that one or more of the sites mutated in the Mrc1-8D variant are indeed phosphorylated in cells, in a Rad53-dependent manner, in response to DNA replication stress.

Apologies – we realise that we neglected to include some important references related to previous phosphoproteomic studies. Three of the eight sites in Mrc1^8D^ (S911, S918, and S920) have been detected by proteomics from cells treated with MMS (Lanz et al., 2021) showing they are indeed phosphorylated in response to replication stress. Another phosphoproteomics screen identified nearby residues (within the C-terminal region we identified) that were also dependent on the presence of Rad53 (Lao et al., 2018). We have added this to the “Mrc1^8D^ slows fork rate in vitro and in vivo and partially rescues rad53 mutant sensitivity” section on page 11. It would be very difficult/impossible to get the kind of mass spec coverage we would need to validate all of the sites, even in vitro.

(3) In addition to the viability assays shown in Figure 6, it would be useful if the authors could provide an assay that monitors replication fork progression to ensure that the Mrc1-8D protein restrains fork progression even in the absence of Rad53.

Addressed in Essential revisions.

Reviewer #3 (Recommendations for the authors):I would really appreciate a figure showing the locations of the 8/14/19 mutations. I feel these should be added to the diagram in S3B.

We have now added figures with the location of the mutations to new Figure 5 —figure supplement 1B.

I suggest the authors construct and test an 8A version. If this allele exhibits more limited effects than 14A, it will strengthen the inference that the 8D changes are not just breaking the protein.

Because the Mrc1^14A^ mutant is nearly wild-type in its ability to promote replication speed in vitro, it would be very surprising if the Mrc1^8A^ mutant, which is effectively a subset of the 14A mutant wasn’t also like wild type. Because we thought it would be highly unlikely the 8A mutant would work less well than the 14A mutant and because of time constraints we decided not to test this mutant.

I think a bit more in vivo analysis of these mutants would be useful and appreciated, especially by experts such as bulk analysis of DNA content during S phase progression. One would predict that the 8D allele would cause slow replication even without drugs. However, it might have little effect given the apparently intact checkpoint signaling and presumably intact origin firing control.

The reviewer makes a great suggestion to look at replication fork rate in the Mrc1^8D^ strains. Because S phase progression by FACS analysis reflects both the state of origin firing and replication fork speed, we instead used a system using BrdU incorporation and replication intermediate size measurements to determine replication fork rates in these cells. This experiment is described in Essential revisions.

One set of easy experiments that seems to be missing is more analysis of the 14A and 19A alleles in vivo. These alleles might be predicted to exhibit sensitivity to HU and MMS. Should effects be noted, it will probably be important to examine Rad53 activation in these mutants similar to analyses in 6B and C to rule out possible effects on transducing the signal to activate Rad53 as opposed to defects in receiving effector signals from activated Rad53. It is possible that the 14A and 19A mutations will have no sensitivities due to incomplete penetrance or redundant mechanisms, but worth a try.

We tested the sensitivity of the Mrc1^19A^ mutant in response to hydroxyurea and didn’t detect any phenotype (see below and new Figure 5 —figure supplement 1C). As the reviewer points out, this could be due to incomplete penetrance (as seen in in vitro replication with Mrc1^19A^ in Figure 5E) or redundant mechanisms. We have added this to the section “Identification of Rad53 phosphorylation sites in Mrc1” on page 10.